# HUMAN-ALIGNED CHESS WITH A BIT OF SEARCH

**Yiming Zhang**[1]  **Athul Paul Jacob**[2]  **Vivian Lai**[3]  **Daniel Fried**[1]  **Daphne Ippolito**[1]
[1]Carnegie Mellon University  [2]MIT  [3]Visa Research

## ABSTRACT

Chess has long been a testbed for AI's quest to match human intelligence, and in recent years, chess AI systems have surpassed the strongest humans at the game. However, these strong AI systems are *not human-aligned*; they are unable to match the skill levels of all human partners or model human-like behaviors beyond piece movement. In this paper, we introduce ALLIE, a chess-playing AI designed to bridge the gap between artificial and human intelligence in this classic game. ALLIE is trained on log sequences of real chess games to model the behaviors of human chess players across the skill spectrum, including non-move behaviors such as pondering times and resignations In offline evaluations, we find that ALLIE exhibits humanlike behavior: it outperforms the existing state-of-the-art in human chess move prediction and "ponders" at critical positions. The model learns to reliably assign reward at each game state, which can be used at inference as a reward function in a novel *time-adaptive* Monte-Carlo tree search (MCTS) procedure, where the amount of search depends on how long humans would think in the same positions. Adaptive search enables remarkable *skill calibration*; in a large-scale online evaluation against players with ratings from 1000 to 2600 Elo, our adaptive search method leads to a skill gap of only 49 Elo on average, substantially outperforming search-free and standard MCTS baselines. Against grandmaster-level (2500 Elo) opponents, ALLIE with adaptive search exhibits the strength of a fellow grandmaster, all while learning *exclusively from humans*.[1]

## 1 INTRODUCTION

Computer chess is among the most studied problems in Artificial Intelligence. In the pursuit of stronger chess engines, decades of hardware and algorithmic improvements since the first computer chess programs (Turing, 1948; Shannon, 1950) have led to the development of increasingly strong chess engines (Campbell et al., 2002). Current state-of-the-art systems, such as Stockfish (Romstad et al., 2008) and AlphaZero (Silver et al., 2017) have reached strength far beyond even the best human players. However, these systems are not calibrated to play at levels matching human strength, and they make moves that are inscrutable even to human experts.

In this work, we revisit the classic challenge of computer chess, but with a different objective: developing a *skill-calibrated* and *human-aligned* chess AI. By *skill-calibrated*, we mean an system that is evenly matched (i.e., winning approximately 50% of games) against players across the skill spectrum, while maintaining humanlike play. Skill calibration of AI systems is an open research challenge, and could prove valuable for domains requiring *superhuman* AI systems to collaborate with and be overseen by imperfect human partners. Similar to McIlroy-Young et al. (2020), we define *human-aligned* as whether the model behaves indistinguishably from a human player. Our definition extends beyond just move selection: other key aspects, such as time spent pondering a move and the decision to resign in losing positions, are fundamental to how humans play chess. By incorporating these humanlike behaviors, our chess engine ALLIE aims to serve as an engaging and realistic sparring partner for human players.[2]

Our approach draws upon recent success in language modeling. Large decoder-only Transformer models, when trained on vast text corpora (Radford et al., 2019; Touvron et al., 2023), learn to

---

[1]Code, data and model weights are available on GitHub.

[2]Pondering in chess means spending time to make a move — humans usually spend more time at critical positions. Resignation is the act of conceding a game out of respect for the other player in a losing position.

generate text that is sometimes indistinguishable to human-written content (Dugan et al., 2023). Similar to language, chess has a natural sequential representation—with moves taking the place of tokens. It is therefore natural to model chess like language: we train a decoder-only Transformer model (Vaswani et al., 2017) on a large dataset of human chess game trajectories to model how humans play chess. Our resulting model predicts human moves at a state-of-the-art level (competitive with GPT-3.5 despite many fewer parameters), and demonstrates humanlike behavior in other aspects of chess play, such as pondering and resigning, which previous systems are incapable of modeling. The model also demonstrates a remarkable ability to predict game outcomes at intermediate board positions, achieved solely through supervision on human game outcomes.

Using ALLIE's next move distribution and value estimation *learned exclusively from humans*, we add *a bit of* search at inference time. Specifically, our *time-adaptive* Monte-Carlo tree search (MCTS) method allocates limited inference budget proportional to the predicted human ponder time, enabling more intensive search at critical positions. In a large-scale human study of 7,483 games with 2,412 human players, we find that our adaptive search method enables skill calibration to strengths ranging from beginner to expert levels with a skill gap of only 49 Elo points on average across the skill spectrum. Against 2500 Elo opponents, our adaptive search method enables ALLIE to achieve near-perfect skill calibration, substantially outperforming both search-free baselines and a traditional MCTS approach with equal computational budget.[3]

## 2 RELATED WORK

Most existing approaches in chess engine development have focused on creating the *best possible* system. Early successful engines like Deep Blue relied on hand-coded rules and extensive search algorithms (Campbell et al., 2002). In contrast, AlphaZero (Silver et al., 2017) used self-play and Monte-Carlo tree search (MCTS) to learn a probability distribution over actions (*policy*) and estimate game outcomes with a *value function*. AlphaZero also employed MCTS at inference time to select winning moves. We explore a variation of this MCTS algorithm in Section 3.3, using policy and value functions learned directly from human games, and inference time search budget allocated proportional to human ponder time.

More recently, McIlroy-Young et al. (2020) introduced 'MAIA', a neural network trained on human chess games rather than through self-play, proposing a new goal of creating a human-aligned chess AI and achieved remarkable accuracy in modeling how humans play chess. Following the success of MAIA, it has been shown chess players can be reliably identified using a small number of games through their playing style (McIlroy-Young et al., 2021), and fine-tuning on individual gameplay substantially boosts the model's capability of predicting the individual's moves (McIlroy-Young et al., 2022). Recently, Maia-2 (Tang et al., 2024) further unifies the Maia models at different skill levels into a single model. Jacob et al. (2022) showed that policy and value functions learned from humans can be combined with MCTS to improve policy strength, and we extend upon their work and demonstrate that adaptive search enables ALLIE to almost perfectly match the strengths of human players up to the grandmaster level. By learning value estimates generated by an oracle search engine, Ruoss et al. (2024) showed that neural networks can achieve grandmaster-level performance without inference-time search. Our approach differs in that our networks are supervised on human data *alone*.

Our proposed method is inspired by Toshniwal et al. (2022)'s idea of treating chess like a language modeling task. Feng et al. (2023) fine-tuned a language model on chess games, books and commentary and demonstrated that the model can track pieces throughout games and solve chess puzzles, and Karvonen (2024) demonstrated that a language model trained to predict chess moves exhibits emergent understanding of chess concepts. Zhang et al. (2024) similarly showed that a Transformer model trained on human games can be made to play at a higher skill level than the games in its training data by using a low sampling temperature.

## 3 BUILDING ALLIE, A HUMAN-ALIGNED CHESS MODEL

Here, we describe how we represent a chess game, and our training and inference methods.

---

[3]Elo is a standard measure of strength in two-player games (higher is stronger). A 2500 Elo level corresponds to 99.6% percentile of players on the popular chess website Lichess.

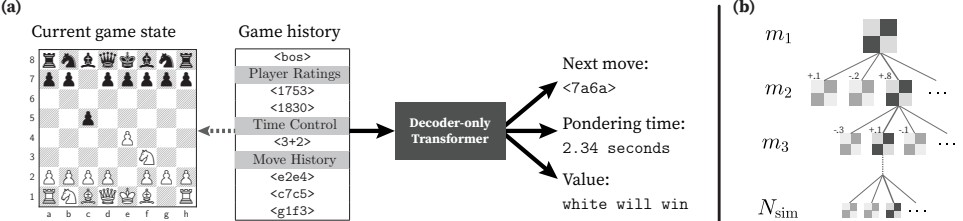

Figure 1: **(a)** The current game state can be represented as the sequence of moves that produced it. This sequence, which also includes metadata on the players' skill and the time setting (e.g. a blitz game), is inputted to a Transformer, which predicts the next move, pondering time for this move, and a value assessment of the move. **(b)** At inference time, we employee Monte-Carlo Tree Search with the value predictions from the model. The number of rollouts $N_{\text{sim}}$ is chosen dynamically based on the predicted pondering time.

### 3.1 REPRESENTING A CHESS GAME SEQUENTIALLY

**Vocabulary** To apply language modeling techniques to chess, we need a sequential representation of a chess game. To this end, we view a chess game as a sequence of moves. We encode moves using Universal Chess Interface (UCI) notation, which specifies every chess move as its starting and ending square (see example in Figure 1). We initialize the language model's vocabulary $\Sigma$ as the set of possible moves under UCI notation (1968 in total). A board state is implied by the sequence of moves that led to that board state. Game metadata, including the two players' skill levels, time control (how much time each player is allowed to take over all the moves in a game), and a termination condition (e.g., whether the game ends with a resignation or checkmate) are added to the vocabulary as special tokens.[4] This representation is compact for training: contextualized by the previous tokens in a sequence, each token in the dataset implicitly maps to a single board state, making training on a dataset with billions of chess positions feasible and efficient.

**Strength conditioning** Player skill in chess is computed using the Elo rating system (Elo, 1967). Elo scores normally fall in the range of 500 (beginner players) to 3000 (top chess professionals). To calibrate the playing strength of ALLIE to different levels of players, we model gameplay under a conditional generation framework (Keskar et al., 2019), where encodings of the Elo ratings of both players are prepended to the game sequence.

The obvious way to encode Elo ratings as tokens would be to add items to our vocabulary representing each Elo score between 500 and 3000. However, this approach runs into data sparsity issues (a small number of games for each individual Elo rating), and this discrete representation fails to encode the fact that scalar distances between Elo scores are meaningful (a difference of 5 between two players' Elo ratings indicates they are much closer in ability than a difference of 500). To address these issues, we introduce *soft* control tokens, which interpolate between a *weak* token, representing 500 Elo, and a *strong* token, representing 3000 Elo. For a player with Elo rating $k$, we compute a soft token $e_k$ by linearly interpolating between the weak and strong tokens: $e_k = \gamma e_{\text{weak}} + (1 - \gamma)e_{\text{strong}}$, where $\gamma = \frac{3000-k}{2500}$. During training, we prefix each game with two soft tokens corresponding to the two players' strengths.

### 3.2 TRAINING ALLIE TO MOVE, PONDER AND EVALUATE

Using a sequential representation of a chess game, we can naturally apply standard sequence modeling techniques to model how human players make moves and when they decide to resign (we treat "resignation" as just another move token the model can assign probability to). ALLIE is built using a decoder-only Transformer model (architecture details in Section 4.2) which inputs the game history as a sequence and has three output heads: (1) a policy head $p_\theta$ that outputs a probability distribution over possible next moves, (2) a pondering-time head $t_\theta$ that outputs the number of seconds a human player would take to come up with this move, and (3) a value assessment head $v_\theta$ that outputs a scalar

---

[4]Time control and skill level are prepended to the start of the game sequence, and termination condition tokens are appended to the end of the game sequence.

value representing who is expected to win the game. The pondering-time and value assessment heads are crucial for the *human-aligned* chess play that we aim to capture. The former allows ALLIE to behave like a human, taking more time to make decisions in complex game states than simple ones, and the latter allows the model to discriminate between good moves and blunders. All three heads combined enable the adaptive MCTS procedure, detailed in Section 3.3.

All three prediction heads are individually parameterized as linear layers applied to the outputs of the final decoder layer. Given a dataset $\mathcal{D} = \{(\mathbf{m}, \mathbf{t}, v)\}$ of chess games, each represented as a sequence of moves $\mathbf{m} \in \Sigma^N$, human think time before each move $\mathbf{t} \in \mathbb{R}^N$ and the ultimate game outcome $v \in \{-1[black\ wins], 0[draw], 1[white\ wins]\}$, we train the model to minimize the log likelihood of the next move and mean squared errors of time and value predictions:

$$\mathcal{L}(\theta) = \sum_{(\mathbf{m},\mathbf{t},v)\in\mathcal{D}} \left( \sum_{1 \leq i \leq N} \left( -\log p_\theta(m_i \mid \mathbf{m}_{<i}) + (t_\theta(\mathbf{m}_{<i}) - t_i)^2 + (v_\theta(\mathbf{m}_{<i}) - v)^2 \right) \right).$$

A similar objective of jointly learning policy and value can be found in MCTS-based reinforcement learning algorithms (Silver et al., 2017; Schrittwieser et al., 2020).

### 3.3 POLICY IMPROVEMENT UNDER TIME CONSTRAINTS WITH ADAPTIVE SEARCH

Virtually all strong chess engines (Romstad et al., 2008; Pascutto & Linscott, 2019) rely on *search*, a process of exploring possible future moves to pick the best move. Past work has shown that search is crucial for achieving strong gameplay (Silver et al., 2017; Jones, 2021). Since ALLIE produces both policy and value estimators, planning algorithms such as Monte-Carlo tree search (MCTS) (Coulom, 2007) can be applied off-the-shelf for policy improvement. As shown in Figure 1b, MCTS works by rolling out multiple moves into the future, selecting paths that are most likely to lead to a win.

State-of-the-art search-based chess engines such as AlphaZero use a constant number of rollout steps for each move, leading to them assessing tens of thousands to millions of positions before playing a move. Such large amounts of search are incompatible with our goal of human-alignment; in blitz games, humans frequently makes moves with <1 second of time usage, and it is practically infeasible to search through such a large number of rollouts on consumer hardware in this timeframe. On the other hand, in critical game states where the model predicts a human would spend more time to ponder, it is plausible that running deeper simulations would allow for better modeling of the elevated depth of human reasoning in such positions and improve policy strength.

To this end, we propose a time-adaptive MCTS procedure that aligns MCTS with human reasoning: at each position $\mathbf{m}$, we dynamically set the number of rollouts $N_{\text{sim}} = \lfloor c \cdot t_\theta(\mathbf{m}) \rfloor$, where $t_\theta(\mathbf{m})$ is the predicted human pondering-time at the position $\mathbf{m}$ and $c$ a constant.[5] Another alternative implementation of time-adaptive MCTS would be to keep searching until a timeout is reached, but we opted against doing this in order to make our implementation independent of hardware efficiency.

## 4 EXPERIMENTAL SETUP

### 4.1 DATASET

We constructed a raw dataset of chess games using all blitz[6] games played in 2022 on Lichess, a popular online chess platform.[7] To address the data's skew toward low-skill-level games, we downsampled the dataset to have roughly equal numbers of games in bins in increments of 100 Elo. From this downsampled dataset, we use 18 thousand games for testing, and the remaining games for training and validation. In total, the training set contains 91 million games and 6.6 billion tokens.

Our primary automatic evaluation metric is move-matching accuracy—how often does the model correctly predict the next move in the game. Following McIlroy-Young et al. (2020), when evaluating accuracy, we discard the first 5 moves of each game, which reduces the impact of opening

---

[5]The value of $c$ is set so that $N_{\text{sim}} = 50$ for the average position. Our MCTS implementation and hyperparameters follow AlphaZero (Silver et al., 2017). See Appendix E.2.

[6]A blitz game is one where each player usually can take 3-5 minutes across all their moves.

[7]https://database.lichess.org/

memorization (there are only so many ways to begin a chess game). We further omit from evaluation any moves made under time pressure (when there is less than 30 seconds on the clock) to avoid the influence of random moves made due to being low on time. This leaves us with 884,049 positions from an evaluation test set. To further evaluate the abilities of ALLIE to produce valid chess moves under *out-of-distribution* game states, we also constructed a dataset of *random* chess games, where each game contains moves that are randomly sampled among legal moves in each position.

## 4.2 MODEL ARCHITECTURE

Our model uses a standard decoder-only Transformer architecture (Vaswani et al., 2017) with 355M parameters. We initialize model parameters (excluding embeddings) using weights from the pre-trained GPT-2 medium model (Radford et al., 2019), and embeddings are trained from scratch since the vocabulary is not shared with natural language. It may seem surprising that that learned model weights for language modeling are useful for a non-linguistic task like chess, but this transfer technique is shown effective in other domains (Papadimitriou & Jurafsky, 2020; Shen et al., 2023). The value prediction head is followed by a tanh activation layer that squeezes the value prediction to the range $[-1, 1]$, with the extreme values corresponding to wins for each of the two players. Time prediction labels are normalized to have variance 1, and all three loss terms are weighted equally. The model is trained for 2M steps with a global batch size of 131,072 tokens on our training set. This corresponds to roughly 40 epochs over the training data. Additional training details and hyperparameters are provided in Appendix E.1. In Appendix F, we explore the effect of both dataset size and parameter count on model capability. We find that our setting is mostly *data-constrained*— model performance is limited by the number of human chess games available on the Internet—and doubling model size has only a small effect on the model's ability of predicting human moves.

## 4.3 BASELINES

We compare our ALLIE's learned policy against MAIA (McIlroy-Young et al., 2020), which, like ALLIE, is trained on human-games to make next-move predictions. MAIA is a family of nine individual models, each trained on Lichess games from players with Elo ratings in a given range. We refer to these as MAIA-$\{1100, 1200, \dots, 1900\}$. The MAIA network architecture is a residual CNN, and their move prediction objective used during training is similar to our approach, but the input representation is board state without full move history information. To unify the different Maia models into a single strong baseline, we define a MAIA* model by adaptively choosing the Maia model with the closest Elo rating to the players' ratings. For example, a 1480-rated game would be evaluated using the Maia-1500 model. We note that publicly available MAIA models are much smaller than ALLIE, and this has an effect on the relative performance of the models. We explore a variant of ALLIE with half the parameters in our ablation study (Section F).

The primary comparative metric we use for automatic evaluation is move-matching accuracy: what fraction of the time does the system correctly predict the move a human would have made. Other aspects of *human-aligned* chess play (e.g., modeling human moves vs. time usage) require different evaluation metrics, which we detail in Section 5. To the best of our knowledge, there are no existing chess engines that model how humans play chess in terms of pondering and resigning, so we do not have a direct comparison with a baseline system for these behaviors.

Though large language models (LLMs) such as OpenAI's GPT-3.5(-turbo-instruct) have not (to our knowledge) been explicitly trained to play chess, they have been shown to reliably produce humanlike next moves.[8] This is accomplished by prompting the LLM with a textual representation of the game state using PGN notation.[9] Due to dependency on the textual PGN notation, this approach is not compatible with OpenAI's latest chat-based LLMs (e.g., GPT-4), and we report prompts and implementation details in Appendix B. It is difficult to make a fair comparison between ALLIE and GPT-3.5 because on the one hand, GPT-3.5 has many more parameters and potentially observed much more chess data during pre-training. On the other hand, GPT-3.5 was never intended to play chess, and the fact that it can play chess is somewhat remarkable. We report GPT-3.5 results just to provide context on performance achievable by a frontier large language model.

---

[8] https://nicholas.carlini.com/writing/2023/chess-llm.html

[9] The Portable Game Notation (PGN) is a popular human-readable and human-writable textual notation for chess games.

Table 1: All configurations of our chess-engine, ALLIE.

| Config. | Description |
|---|---|
| ALLIE-POLICY | Softmax sampling according to $p_\theta$ with unit temperature. |
| ALLIE-GREEDY | Greedy decoding according to $p_\theta$ conditioned on a 2,500 Elo level. |
| ALLIE-SEARCH | ALLIE-POLICY with non-adaptive MCTS (50 rollouts). |
| ALLIE-ADAPTIVE-SEARCH | ALLIE-POLICY with adaptive MCTS ($c$ set such that MCTS performs 50 rollouts on average across all positions). |

Table 2: ALLIE *learns to play valid chess moves.* 95% confidence intervals are shown.

| Evaluation set | Top-1 move is valid (%) |
|---|---|
| Lichess | $100.0 \pm 0.0$ |
| Lichess (under *check*) | $100.0 \pm 0.0$ |
| Random | $99.9 \pm 0.0$ |
| Random (under *check*) | $96.6 \pm 0.0$ |

Table 3: ALLIE-POLICY *outperforms state-of-the-art methods in human move prediction.* Move prediction accuracy with 95% confidence intervals are reported in the table.

| Human plays... | ALLIE (%) | Maia⋆ (%) | GPT-3.5 (%) |
|---|---|---|---|
| All moves | $55.7 \pm 0.1$ | $51.6 \pm 0.1$ | $53.7 \pm 0.1$ |
| *Castling* | $74.3 \pm 0.5$ | $73.3 \pm 0.6$ | $72.4 \pm 0.6$ |
| *En passant* | $70.4 \pm 4.1$ | $67.7 \pm 4.2$ | $71.4 \pm 4.0$ |
| *Pawn promotion* | $86.9 \pm 1.7$ | $85.1 \pm 1.8$ | $86.0 \pm 1.7$ |
| *Threefold repetition* | $92.0 \pm 4.6$ | $87.0 \pm 5.7$ | $92.8 \pm 4.4$ |

## 4.4 LARGE-SCALE HUMAN STUDY

In addition to conducting offline evaluation, we deployed the four configurations of ALLIE described in Table 1 as well as MAIA⋆, to play blitz games on the website Lichess. ALLIE-POLICY was conditioned to play adaptively at the opponent's strength, and moves were sampled from the model distribution $p_\theta$. ALLIE-GREEDY was conditioned to play at a 2,500 skill level, and top moves under the model distribution are played. This setting allowed us to measure the upper bound of the policy strength.[10] ALLIE-SEARCH and ALLIE-ADAPTIVE-SEARCH employ inference-time MCTS to improve move selection, with the latter using an adaptive number of rollouts. Overall, we collected 7,483 blitz games with 2,412 human players over a multi-week period. After each game, players were invited to fill out a survey about their experience. Survey results can be found in Appendix D.2.

## 5 RESULTS

To apply inference-time search to ALLIE, we first need to understand if chess is at all learnable from human-generated data (Section 5.1), and if so, how well ALLIE models human gameplay (Section 5.2). We discuss our main results on adaptive MCTS and skill calibration in a large-scale study against human players in Section 5.3.

## 5.1 DOES ALLIE LEARN THE RULES OF CHESS?

First, we ask whether the rules of chess are learnable from human-generated chess data. The model produces a softmax distribution over roughly two thousand possible chess moves, and we can test if the model has indeed learned the rules of chess by checking if model assigns high probability to valid moves, and low probability to invalid moves. While we evaluate the model's behavior on actual human games, it is also important to test if the model can generalize to *out-of-distribution* positions that are rare in human games but are nevertheless valid: a model that has learned the rules of chess should play legal moves in randomly generated games as well. Beyond testing the model behavior in the aggregate, we further examine the model's behavior when special chess rules restricting valid moves (e.g., *check*) are in effect.[11]

In Table 2, we report how often the top move from the model distribution is valid. On both the human and random evaluation sets, we find that the top move is *almost always* valid: 100% of the time on human games, and 99.9% of the time on random games. Softmax distributions by definition assign

---

[10] A rating of 2500 is typically considered as the threshold for grandmaster level play.
[11] See Appendix A for a glossary of chess terms.

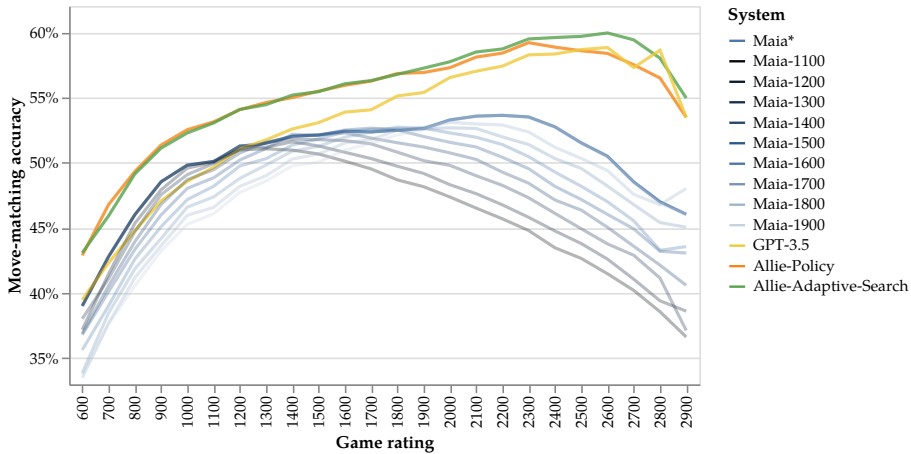

Figure 2: *Adaptive search enables matching human moves at expert levels.* Move-matching accuracy of ALLIE-POLICY, ALLIE-ADAPTIVE-SEARCH, MAIA and GPT-3.5 are reported across skill levels. ALLIE-SEARCH has virtually the same move matching accuracy as ALLIE-ADAPTIVE-SEARCH and is omitted from the figure.

non-zero probabilities to all (including invalid) moves, but this probability is vanishingly small: 0.2% in both human and random games (see Table 7 in Appendix C.1). In positions where the king is under *check*,[12] the model still only assigns 0.2% of probability to all invalid moves. Our results suggest that the model has indeed learned the rules of chess from observing human chess games, and generalizes reasonably well to out-of-distribution positions.

## 5.2    HOW WELL DOES ALLIE MODEL HUMAN GAMEPLAY?

The ideal human-aligned chess bot should behave indistinguishably from a human chess player. A major aspect of humanlikeness is in the moves played: for a given game state, a humanlike chess bot should play the same move as a human would in the same position. Beyond moves played, we argue that it is important to match the time humans ponder their moves before taking them, and resign when appropriate—these are also essential components of how humans play chess.

**Moves.**    On the Lichess evaluation set, we compare how often ALLIE, GPT-3.5, and the MAIA models play the same moves as humans. Following McIlroy-Young et al. (2020), we consider the *move-matching accuracy* metric, defined as the fraction of top-1 moves under the model distribution that matches human moves at the same positions. Over the entire test set, the top move produced by ALLIE matches human moves 55.7% of the time, compared to MAIA*'s 51.6% and GPT-3.5's 53.7% (Table 3).Shown in Figure 2, we find that ALLIE matches human moves more accurately than MAIA and GPT-3.5 models across almost the entire skill spectrum. Notably, ALLIE-ADAPTIVE-SEARCH outperforms ALLIE-POLICY at 2300 Elo and above, providing evidence that search is crucial for modeling the behavior of expert-level human players (Jacob et al., 2022).

We further report move-matching accuracy of special moves such as *castling*, *en passant*, *pawn promotion*, and *threefold repetition* in Table 3. ALLIE reaches higher move-matching accuracy than MAIA* for all four types of special moves, and is competitive with GPT-3.5 overall.

**Pondering time and resignation.**    Additional dimensions of human behavior, including pondering time and resignation, are also key aspects in humanlike gameplay. We find a strong correlation between the model's predicted think time and human think time, with Pearson's $r = 0.697$. This suggests that ALLIE successfully learns to predict when humans do and do not ponder in a position. Figure 3 shows the distribution of ALLIE's predicted think time for different amounts of time spent by humans. There is a clear monotonic relationship, but interestingly ALLIE tends to predict lower

---

[12]This is a game state where the set of valid moves is more restricted than usual; the player must make a move that prevents the opponent from capturing their king piece.

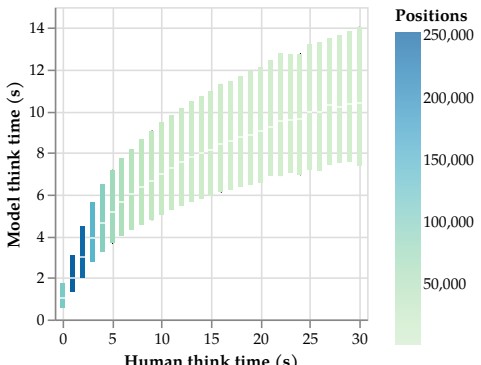

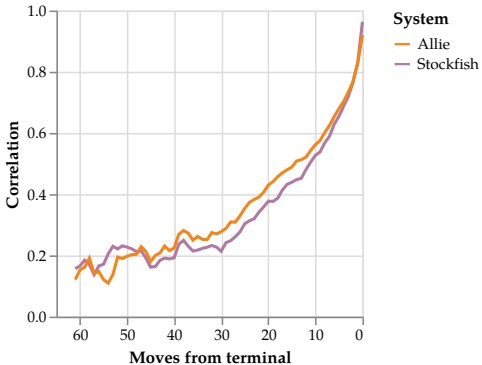

Figure 3: ALLIE*'s time predictions are strongly correlated with ground-truth human time usage.* In the figure, we show median and interquartile range of ALLIE's predicted think time for different amount of time spent by humans and observe a clear monotonic relationship.

Figure 4: ALLIE *learns to assign reliable value estimates to board states by observing game outcomes alone.* We report Pearson's $r$ correlation of value estimates by ALLIE and Stockfish with game outcomes. Game outcomes are increasingly predictable as the game progresses.

pondering times than humans do. This is probably because of the skew in pondering time distribution: the majority of moves in blitz games is played under 5 seconds, and the model is incentivized to "hedge" its prediction and output shorter pondering times.

We further evaluate whether ALLIE can resign in losing positions like humans. We define resignation as when a special resignation token `<resign>` is assigned higher likelihood than all valid moves on the board, and the predicted board value is below -0.9 from the perspective of ALLIE. We focus our analysis on both the true positive rate (TPR), i.e., the number of positions where the model resigns when humans resign, and false positive rate (FPR), i.e., the number of positions where the model resigns when humans do not resign. Over the evaluation set, we observe a TPR of 86.4%, indicating ALLIE usually resigns when a human would. ALLIE almost never resigns when a human wouldn't, with a FPR of 0.1%. Our results highlight that ALLIE models human chess play holistically, not only in terms of moves played, but also in pondering time and resignation when approriate.

**Reliable board value estimate.** Before applying a search algorithm such as Monte-Carlo tree search (MCTS), we need a value function that guides exploration of promising game states. Recall that ALLIE is trained to predict the outcome of games at each position—which can be conveniently interpreted as a board value function. In Figure 4, we show how well ALLIE's value function and an oracle value function correlate with game outcomes.[13] By observing only outcomes of games without additional supervision, we find that ALLIE learns to assign surprisingly reliable value estimates to chess board states: ALLIE's value estimates closely match that of the oracle, and predicts game outcomes just as well. Notably, ALLIE has access to game metadata (in particular, player skill levels) that Stockfish does not, which may explain why it even outperforms Stockfish sometimes. Our results suggest that ALLIE learns credit assignment in chess by observing game outcomes alone, and provides the foundation for applying value-guided search methods such as MCTS.

### 5.3 EVALUATING SKILL CALIBRATION VIA GAMES WITH HUMANS

Our offline evaluations suggest that ALLIE predicts human behavior well, but to study whether ALLIE could calibrate to strength of human players, we had ALLIE play against real humans at a variety of skill levels. A chess engine that is perfectly *skill-calibrated* should win 50% of games against players regardless of their skill level. Inspired by the expected calibration error metric (Naeini et al., 2015; Guo et al., 2017), we define a *skill calibration error* (SCE) metric. Games between the chess engine and humans are first partitioned into equally spaced bins based on skill level (player Elo). For a bin of games $B$ between the evaluated system and human players, we take the absolute difference between the system's estimated performance on the set of games, and the average Elo of the human

---

[13]We use evaluations of Stockfish (Romstad et al., 2008) after $10^6$ nodes searched.

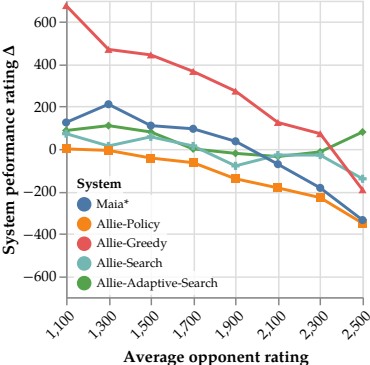

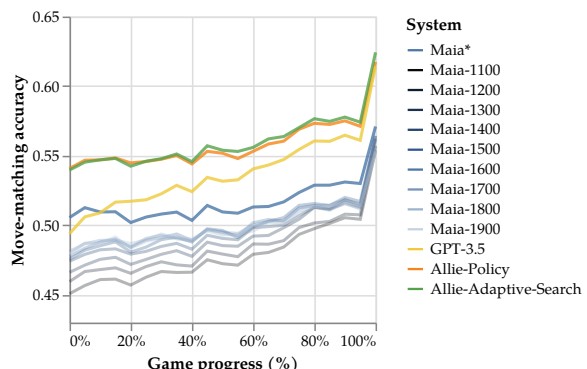

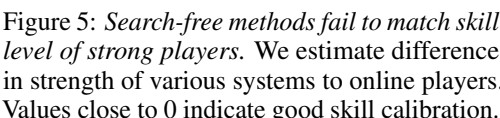

Figure 5: *Search-free methods fail to match skill level of strong players.* We estimate difference in strength of various systems to online players. Values close to 0 indicate good skill calibration.

Figure 6: Human move prediction accuracy increases consistently across models as the game progresses and jumps up sharply in the last portion of the game.

Table 4: *Adaptive search enables remarkable skill calibration.* Mean and maximum skill calibration errors are computed by binning human players into 200-Elo groups. We also report systems' estimated performance against players at the lower and upper Elo ends of the skill spectrum.

| | *Skill Calibration Error* | | *Online performance vs. ...* | |
| System | Mean ↓ | Max ↓ | 1100-rated | 2500-rated |
|---|---|---|---|---|
| *Search-free* | | | | |
| MAIA* | 146 | 336 | 1251 | 2138 |
| ALLIE-POLICY | 127 | 351 | 1134 | 2136 |
| ALLIE-GREEDY | 328 | 677 | 1799 | 2260 |
| *Search-based* | | | | |
| ALLIE-SEARCH | 80 | 166 | 1180 | 2318 |
| ALLIE-ADAPTIVE-SEARCH | 49 | 95 | 1196 | 2528 |

players as the calibration error:[14]

$$\text{SCE}(B) = |\text{SystemElo}(B) - \text{HumanElo}(B)| \, .$$

**Search-free methods do not match the strength of experts.** In Figure 5, we show estimated ratings of the systems against human players across different strength levels, and well-calibrated systems should have a rating difference close to 0. Mean and maximum skill calibration errors are reported in Table 4. We find that ALLIE-POLICY, ALLIE-GREEDY and MAIA* are *not calibrated* to opponent strength. ALLIE-POLICY and MAIA are more or less evenly matched against players below 2100 Elo, but against players above 2400 Elo, both models perform poorly, with ALLIE-POLICY scoring 11.1% and MAIA* scoring 12.5% on average. ALLIE-GREEDY is considerably stronger than weak players (< 2100 Elo), and yet still loses 75% of games to players above 2400 Elo. All search-free systems perform progressively worse against stronger players, suggesting that strength conditioning, sampling temperature (ALLIE-GREEDY) or multiple expert models for different skill levels (MAIA*) may not be sufficient to match the strength of strong human players.

**Skill-calibrated chess play with adaptive search.** Despite being a strong human move prediction model, search-free ALLIE configurations do not match the level of gameplay of strong (≥ 2000 Elo) players. Qualitatively, models blunder pieces and make suboptimal moves in ways that strong players do not (see online players' feedback in Figure 11, Section D.2). In this section, we discuss how we can improve the skill calibration of ALLIE—in particular its performance against strong players—and maintain humanlike play by incorporating an adaptive search method. Recall that

---

[14]We follow rules of the International Chess Federation for computing performance Elo and report evaluation details in the appendix D.1.

Table 5: Some examples of the qualitative feedback we received in our post-game survey.

| System | Feedback |
|---|---|
| ALLIE-ADAPTIVE-SEARCH | I liked the fact Allie plays like a human, and makes human mistakes. She's not like, let's say, Stockfish level 1 making absurd mistakes, nor an inhuman AI with perfect play, but a humanlike player that fights for a win and makes human-reasonable moves. Honestly, I'm not a top player, but I like to play with similar opponents and I'm also a programmer with interest in AI, and I feel satisfied with Allie's behaviour. Great job :) |
| ALLIE-POLICY | I really felt like I was playing against a human, but I have some opinions on this robot: Firstly, I noticed that he plays the opening well, which is a very good thing Secondly, I also noticed that in the middle of the game his accuracy decreases somewhat, he makes mistakes and inaccurate moves, and this is just like a human. |

ALLIE-ADAPTIVE-SEARCH uses an adaptive search budget allocated linearly according to predicted human pondering time at each position, and we compare it with an equal-compute MCTS baseline, ALLIE-SEARCH.

We find that ALLIE-ADAPTIVE-SEARCH improves skill calibration remarkably, achieving an average skill calibration error of 49 Elo, and a maximum skill calibration error of 95 Elo. Figure 5 helps contextualize this finding, where we see the performance ratings of ALLIE-ADAPTIVE-SEARCH exhibit a near-linear relationship with opponent ratings. This is a substantial improvement over all search-free systems, all of which underperform $\geq$ 2400 Elo players by at least 200 Elo points.

More surprisingly, ALLIE-ADAPTIVE-SEARCH outperforms standard AlphaZero-like MCTS (ALLIE-SEARCH), in both overall skill calibration and performance against 2500 Elo human players. Our findings suggest that humanlike reasoning at "critical" positions is useful for reaching expert-level chess. Crucially, ALLIE-SEARCH and ALLIE-ADAPTIVE-SEARCH maintain humanlike play, both achieving a move-matching accuracy of 55.9% compared to 55.7% for ALLIE-POLICY and 51.6% for MAIA$^\star$.

## 6    DISCUSSION

In this work, we demonstrate a method for training a state-of-the-art chess AI that models how humans play chess: our system ALLIE exhibits remarkable precision in playing humanlike moves, as well as pondering and resigning like humans. Through a time-adaptive Monte-Carlo tree search algorithm, ALLIE can be evenly matched with players from beginner (1100 Elo) to expert level (2500 Elo) with almost no skill gap, by learning chess *exclusively from humans* without the need of distilling from a strong chess engine. We believe the techniques developed in this paper have broad applicability for other settings where aligning AI models with *imperfect* human reasoning is crucial, and we look forward to future explorations in other complex settings, such as the alignment and oversight of superhuman AI systems.

While offline evaluation metrics and quantitative analysis of games with real human players reveal ALLIE's strengths, especially relative to prior approaches, more progress is still necessary to fully realize our goal of a human-aligned chess engine. In qualitative feedback, many players were positive about ALLIE (see Table 5), but several shortcomings were also repeatedly emphasized. Players especially noted ALLIE's propensity toward late-game blunders and that its pondering times were sometimes long in positions where there is only one reasonable move. However, since players all knew they were playing against a bot, it is hard to disentangle their perspectives from this knowledge. For example, contrasting with the qualitative feedback, we empirically observed that move prediction accuracy actually improves as games progress, especially in the last few turns (see Figure 6). For future work, it would be interesting to conduct a proper Turing test, where players do not know whether they are playing against an AI or a human-player of a similar Elo level.

Our approach relies on pre-training, which is limited by available data: the vast majority of online chess games are played at fast time controls, and therefore it is more challenging to use data-driven methods to model human behavior in slower games. Future work should explore methods to model human reasoning in slower games, where players have more time to think and make more accurate moves, and test the generalization of our approach to different time controls and game formats.

ACKNOWLEDGMENTS

We thank CM Ethan Gu and thousands of Lichess players who tested our systems and provided feedback, as well as the Lichess team and contributors for creating a free and open platform for chess. We are grateful to Nicholas Carlini, Harshita Diddee, Mourad Heddaya, Xinyue Liu, Vinay Samuels, Chenhao Tan, Barry Wang, and Xinran Zhao for invaluable discussions. YZ is supported by an OpenAI Superalignment Fellowship. YZ and DI are supported by grants from Cisco and CMU Cylab Seed Funding.

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

# A GLOSSARY OF CHESS TERMS

In this section, we provide a glossary of chess terms that are used throughout the paper. The terms are summarized in Table 6.

Table 6: Chess glossary.

| Chess term | Definition |
|---|---|
| Check | A situation in which a player's king is under direct attack by an opponent's piece. The player must resolve the check on their next move. Check limits the number of valid moves in a position. |
| Castling | A special move involving the king and either rook. The king moves two squares towards the rook, and the rook moves to the square the king crossed. |
| En passant | A special pawn capture that can occur immediately after a pawn makes a double-step move from its starting position. The opposing pawn can capture it as if it had only moved one square. |
| Pawn promotion | When a pawn reaches the opposite end of the board, it can be promoted to any other piece (usually a queen) of the same color, except a king. |
| Threefold repetition | A rule that states a player can claim a draw if the same position occurs three times during a game, with the same player to move each time. |

# B GPT-3.5 EVALUATION

Following the implementation of Carlini (2023), we encode chess move sequences in a PGN format (see Figure 7) and feed them as prompt to GPT-3.5-turbo-instruct for evaluation. Note that we were unable to use the latest OpenAI models like GPT-4 since this evaluation requires access to a non-chat language model API. We use greedy decoding to generate the next move, and in the rare case when the model does not output a legal move, a random move is played.

```
[White "Garry Kasparov"]
[Black "Magnus Carlsen"]
[Result "1/2-1/2"]
[WhiteElo "2900"]
[BlackElo "2800"]

1. e4 e5 2. Nf3
```

Figure 7: Prompt for GPT-3.5-turbo-instruct evaluation.

## C  OFFLINE EVALUATION

### C.1  LEGAL MOVES

In Table 7, we show that ALLIE not only learns to assign high probability to valid moves in human games but also in out-of-distribution, randomly generated games. Under a softmax distribution, the probability mass of all invalid moves is low, indicating that the model is capable of distinguishing between valid and invalid moves.

Table 7: ALLIE learns to play valid chess moves. 95% confidence intervals are shown.

| Evaluation set | Top move is valid (%) | Probability mass of all invalid moves (%) |
|---|---|---|
| Lichess | $100.0 \pm 0.0$ | $0.2 \pm 0.0$ |
| Lichess (under *check*) | $100.0 \pm 0.0$ | $0.2 \pm 0.0$ |
| Random | $99.9 \pm 0.0$ | $0.2 \pm 0.1$ |
| Random (under *check*) | $96.6 \pm 0.0$ | $4.1 \pm 0.2$ |

### C.2  HUMAN MOVE PREDICTION

Overall, we find ALLIE outperform state-of-the-art methods in human move prediction (Table 8). Similar to the findings of Jacob et al. (2022), we find that, adding Monte-Carlo tree search (ALLIE-ADAPTIVE-SEARCH) improves upon a pure imitation learning policy (ALLIE-POLICY). Another interesting observation is that as the game progresses, human moves become increasingly predictable, as shown in Figure 6.

Table 8: ALLIE outperforms state-of-the-art methods in human move prediction. Move prediction accuracy with 95% confidence intervals are reported.

| Human plays... | ALLIE-POLICY (%) | ALLIE-ADAPTIVE-SEARCH (%) | Maia* (%) | GPT-3.5 (%) |
|---|---|---|---|---|
| All moves | $55.7 \pm 0.1$ | $55.9 \pm 0.1$ | $51.6 \pm 0.1$ | $53.7 \pm 0.1$ |
| *Castling* | $74.3 \pm 0.5$ | $74.3 \pm 0.5$ | $73.3 \pm 0.6$ | $72.4 \pm 0.6$ |
| *En passant* | $70.4 \pm 4.1$ | $71.0 \pm 4.0$ | $67.7 \pm 4.2$ | $71.4 \pm 4.0$ |
| *Pawn promotion* | $86.9 \pm 1.7$ | $87.5 \pm 1.6$ | $85.1 \pm 1.8$ | $86.0 \pm 1.7$ |
| *Threefold repetition* | $92.0 \pm 4.6$ | $90.6 \pm 4.9$ | $87.0 \pm 5.7$ | $92.8 \pm 4.4$ |

### C.3  ANALYSIS OF ALLIE'S VALUE PREDICTIONS

ALLIE's supervision on human game outcomes is designed to teach the model to assign high values to positions that a human can convert, as opposed to positions that are theoretically winning under perfect play. This section explores the correlation between ALLIE's value predictions and human game outcomes. We also qualitatively analyze a few positions where ALLIE diverges from Stockfish in its value predictions.

#### C.3.1  ROLE OF METADATA IN VALUE PREDICTION

To investigate the role of player Elo in value prediction, we compared the accuracy of ALLIE's value predictions against Stockfish evaluations in predicting game outcomes at various stages of the game. The analysis was conducted on decisive games, excluding draws.

Table 9: Game outcome prediction accuracy with a $\leq 10$ Elo gap between players

| Game Phase | Stockfish | ALLIE |
|---|---|---|
| Opening | 50.4% | 55.2% |
| Midgame | 62.9% | 65.6% |
| Endgame | 73.9% | 74.3% |

Table 10: Game outcome prediction accuracy with a >100 Elo gap between players

| Game Phase | Stockfish | ALLIE |
|---|---|---|
| Opening | 65.2% | 76.8% |
| Midgame | 69.9% | 79.3% |
| Endgame | 82.5% | 83.7% |

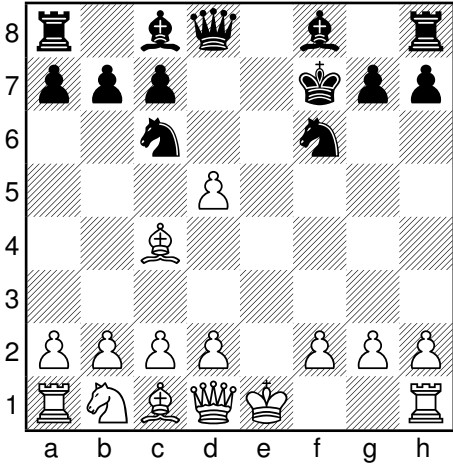

Figure 8: Opening position where value predictions of ALLIE and Stockfish diverge.

Elo provides significant information about the likely winner. In games with at least a 100 Elo gap, ALLIE outperforms Stockfish by 9.4% in the midgame, compared to 2.3% in games with a 10 Elo gap. However, even in games with minimal skill differences, ALLIE consistently outperforms Stockfish across all phases. Taken together, these results suggest that ALLIE's value predictions are more reliable in human games than Stockfish's, even without information about the players' skill level.

### C.3.2 QUALITATIVE ANALYSIS OF POSITION CONVERSION

In this section, we qualitatively analyze several positions from the test set, where ALLIE and Stockfish diverged in the sign of their value predictions.

In an opening position (Figure 8), black has a knight for two pawns but cannot castle. Black is objectively winning (Stockfish assigns a 77.0% probability of winning to black), while ALLIE favors white with a 78.8% probability, since the position requires a series of precise moves to convert. In the actual game, black misplayed the position and lost the game within five moves, suggesting ALLIE may be able to better value positions that are objectively winning but challenging for humans to convert.

In an endgame position (Figure 9), ALLIE surprisingly considers black winning with a 75% probability, despite white having an extra queen (Stockfish assigns 98% to white). ALLIE likely inferred time pressure on white from the move history, as black's predicted advantage began several moves ago when white blundered a rook.

However, ALLIE exhibits blindspots with evaluating checkmates and sacrifices, which strong human players can calculate. In a midgame position (Figure 10), ALLIE gives black a 67% probability of winning, but white is objectively winning with the queen sacrifice Qe7. The human player finds this move and wins, as Stockfish predicted. This inability to predict checkmates and sacrifices is not surprising, since reliable position evaluation *without search* is very challenging.

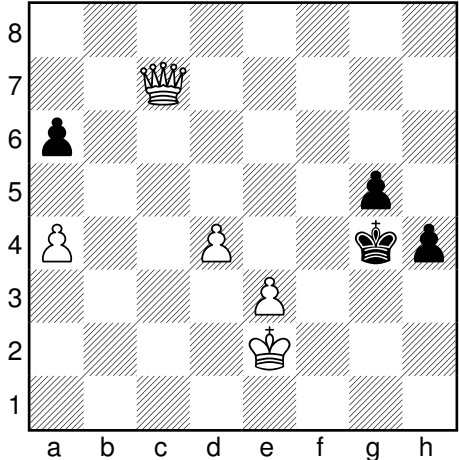

Figure 9: Endgame position where value predictions of ALLIE and Stockfish diverge.

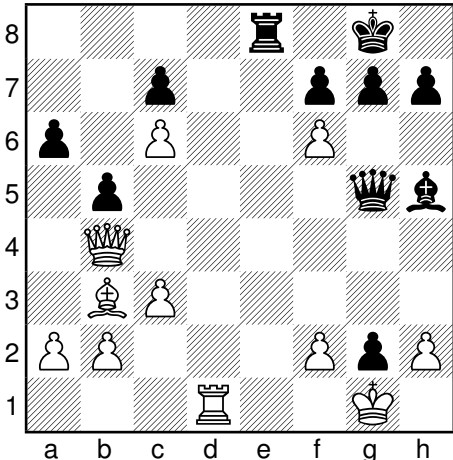

Figure 10: Midgame position where ALLIE ignores a critical queen sacrifice.

# D ONLINE EVALUATION

## D.1 ESTIMATION OF PERFORMANCE ELO

Our estimation of performance Elo ratings follows guidelines of the International Chess Federation (FIDE). Let $r$ denote the average Elo rating of the opponents, and $p$ represent the player's average score against these opponents. FIDE provides a table of estimated rating differences $dp$ corresponding to various values of $p$. For example, if $p = 0.5$, then $dp = 0$, and if $p = 0.75$, then $dp = 193$. These values indicate that a player scoring 50% against their opponents is performing at the same Elo level, while a 75% score suggests a performance 193 Elo points above the opposition's average. The complete table of estimated rating differences can be found in the FIDE Handbook[15]. To calculate the performance rating, one would add the rating difference $dp$ to the average opponent rating $r$. This method provides a standardized approach to estimate a player's performance level based on their results against opponents of known strength.

## D.2 SURVEY RESULTS

In Figure 11, we show the results of a post-game survey where human players were asked to rate the humanlikeness and enjoyability of the systems. We find that ALLIE is rated as more humanlike

---

[15]See https://handbook.fide.com/chapter/B022024 for the full rating difference table.

(28.9% of participants strongly agree) compared to MAIA (24.8%) and more enjoyable to play against (38.6% vs. 27.5%).

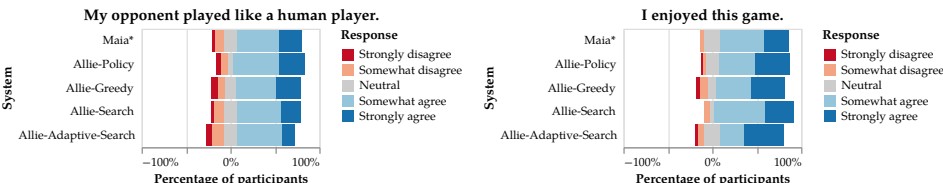

Figure 11: Survey responses.

## D.3 QUALITATIVE FEEDBACK

We provide additional examples of the qualitative feedback we received in our post-game survey in Table 11.

Table 11: Additional examples of the qualitative feedback we received in our post-game survey.

| System | Player Elo | Feedback |
|---|---|---|
| ALLIE-POLICY | 1640 | Played very human-like, resigned at the exact time a human would, and got weaker and sort of "demotivated" as she was losing just like a human. Amazing chess bot |
| ALLIE-POLICY | 1940 | It's very close to being human-like. The thing I will say is that sometimes it appears to take non-obvious moves with no clear "plan" and I have yet to get it to resign. It also seems to be quite a bit weaker than I am, and I don't really play Blitz so I can't imagine I'm very good. |
| ALLIE-POLICY | 2038 | As in the last game I won against Allie, the dropped piece seemed to come out of nowhere. It wasn't a missed tactic or anything like that, but a bad sacrifice. Sometimes of course this will happen against humans, but both of the games I played where this happened to me, it was hard to see any lines (where I didn't outright blunder two or more moves in a row) where the sacrifice would lead to anything. I also expected a resignation at the end. |
| ALLIE-GREEDY | 1139 | Felt human - sometimes when I play Lichess's implementation of Stockfish at a level appropriate for my skill, it makes really bizarre moves, even catastrophic. Maybe they're calculated blunders for noobs like myself, but they're unrealistic. A novice might miss an obvious fork or skewer, but they would never give up their queen for no reason. They would at least try to save it, even if ultimately impossible. Allie doesn't seem to do that. |
| ALLIE-GREEDY | 912 | It seemed at the end that the bot's goal was to clean out my pieces and promote a pawn for a second queen to checkmate rather than just go for a checkmate. (I suppose it's possible that the promotion would have been fewer turns–I'd have to go back and check.) But I feel like a human player would have just gone for a QK v K-style checkmate rather than clean out several of my pawns to make an easy promotion. |
| ALLIE-GREEDY | 2008 | did not take the pawn on e4. Then played what it feels like a pretty accurate series of moves later on in the game. From move 21 the bot played all the best moves some of which feel pretty strong. |
| ALLIE-ADAPTIVE-SEARCH | 1637 | The bot is plays very much like a human. It understood when it had to move fast and when it had to take time. The opening was a little inaccurate but other than that the bot is really good. |
| ALLIE-ADAPTIVE-SEARCH | 1998 | I'd say all moves up until 26. Qc6 were human. Qc6 is slightly unexpected but not that bad.
It was a bit strange that it took a few seconds to take the rook on move 30, because a real human would have understood what they were doing by 29... Be3 and taken immediately to finish the combination.
Just like Maia I don't think it knows what to do in the endgame, which probably contributed to the blunder.
I slightly expected the bot to do 49... Be4 or Ra2 or something to stop the pawn, but no. |
| ALLIE-ADAPTIVE-SEARCH | 2004 | In terms of play I think what I found the least human like was it's willingness to trade when it was down a full piece. My intuition is that these very low level concepts like that even very suboptimal moves being practically better because it increases the long term probability of blunders, is something bots of all strengths struggle with.
However, my opinion is obviously affected by me knowing that I was playing a bot and I'm pretty sure I wouldn't have suspected anything if this was just a normal game! Very cool project! |

# E   TRAINING AND INFERENCE

## E.1   PRE-TRAINING HYPERPARAMETERS

ALLIE is a GPT-2-style (Brown et al., 2020) transformer decoder model with 355M parameters, trained on a dataset of 6.6 billion tokens. We use a global batch size of 131,072 tokens, a learning rate of $6 \times 10^{-4}$, decaying to $1 \times 10^{-5}$ using cosine annealing (Loshchilov & Hutter, 2017), and a maximum sequence length of 512 tokens. The model is trained for 2 million steps, which took approximately 2 weeks on 8 NVIDIA A6000 GPUs using bfloat16 precision.

## E.2   MCTS IMPLEMENTATION DETAILS

Our MCTS implementation and hyperparameters follow a variant of AlphaZero (Silver et al., 2017) proposed by Grill et al. (2020). A way to view MCTS is KL-regularized policy optimization (Grill et al., 2020): in the limit, MCTS produces an optimized policy $\pi$ that maximizes search $Q$ values with KL regularization towards the model policy $p_\theta$ learned from humans:

$$\pi = \arg\max_\pi \sum_a Q(s,a)\pi(s,a) - \lambda D_{\mathrm{KL}}\left(\pi \parallel p_\theta\right). \tag{1}$$

This regularization is key to prevent the search from diverging from the model policy (Jacob et al., 2022), and the KL-regularization strength $\lambda \sim c/\sqrt{N_{\mathrm{sim}}}$, where $c$ is a hyperparameter. In standard MCTS (ALLIE-SEARCH) with fixed number of rollouts, $N_{\mathrm{sim}}$ is fixed, and $\lambda$ is a constant. In adaptive MCTS (ALLIE-ADAPTIVE-SEARCH), we scale $c$ by the square root of the search budget to achieve the same effect of a constant regularization strength. We refer the interested reader to (Silver et al., 2017; Grill et al., 2020) for more details on the MCTS algorithm and its implementation.

# F   ABLATIONS

To assess the impact of the training, data, and model decisions on ALLIE's capability to play humanlike chess, we conduct ablation studies with the following scenarios:

- **Half data**: ALLIE trained on 50% of the dataset for the same number of steps.
- **Half compute**: ALLIE trained on the full dataset for 50% of the steps.
- **Half parameters**: A smaller ALLIE model (124M) with roughly half the parameters, keeping the training data and steps unchanged.
- **Double parameters**: A larger ALLIE model (774M) with roughly double the parameters, keeping the training data and steps unchanged.

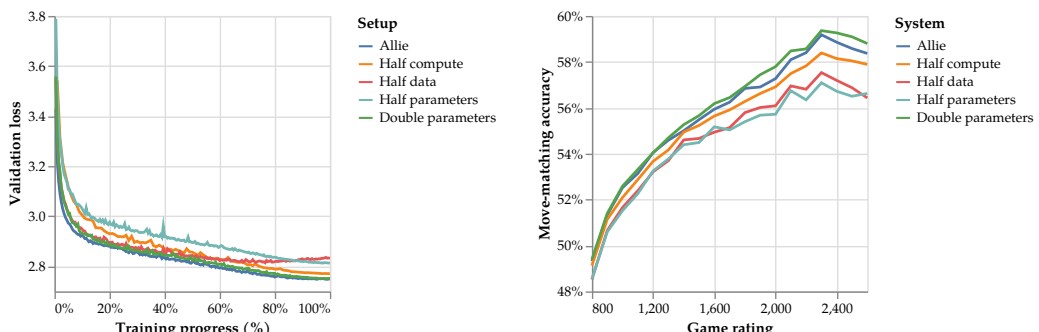

Figure 12: **Left**: Validation loss of ALLIE and ablations throughout training. **Right**: Move-matching accuracy of ALLIE and ablations on the evaluation set.

**Training data and compute.**   We find that the size of the training dataset has a measurable impact on the final loss and move-matching accuracy of the model. Halving the training data leads to a 1.0% decrease in move-matching accuracy over the entire dataset, with signs of overfitting emerging towards the end of training.[16] Conversely, halving the compute (training tokens) minimally affects the final model performance, likely because the model still undergoes approximately 20 epochs of training over the dataset. These observations suggest that the scaling of our training setup is *data-constrained* (Muennighoff et al., 2023), making substantial gains challenging without additional data. Notably, our dataset contains an entire year of blitz games on Lichess, representing a substantial portion of publicly available internet games, thus creating a 10x larger dataset would be difficult.

**Model size.**   Another factor affecting ALLIE's performance is the model size. Halving the model size moderately impacts performance, resulting in a 1.2% decrease in move-matching accuracy. Conversely, doubling the model size yields minimal gains, with only a 0.3% increase in move-matching accuracy. The diminishing returns on model size suggest that further performance improvements through scaling up the model size may be limited without additional data.

We report individual validation losses of ALLIE and ablations throughout training in Figure 13. We find that the language modeling loss and the time prediction loss are stable across ablations and decrease throughout training. Notably when trained on half the data, the model overfits to the value prediction objective towards the end of training.

---

[16]Overfitting is only observed in the value prediction objective (Appendix F).

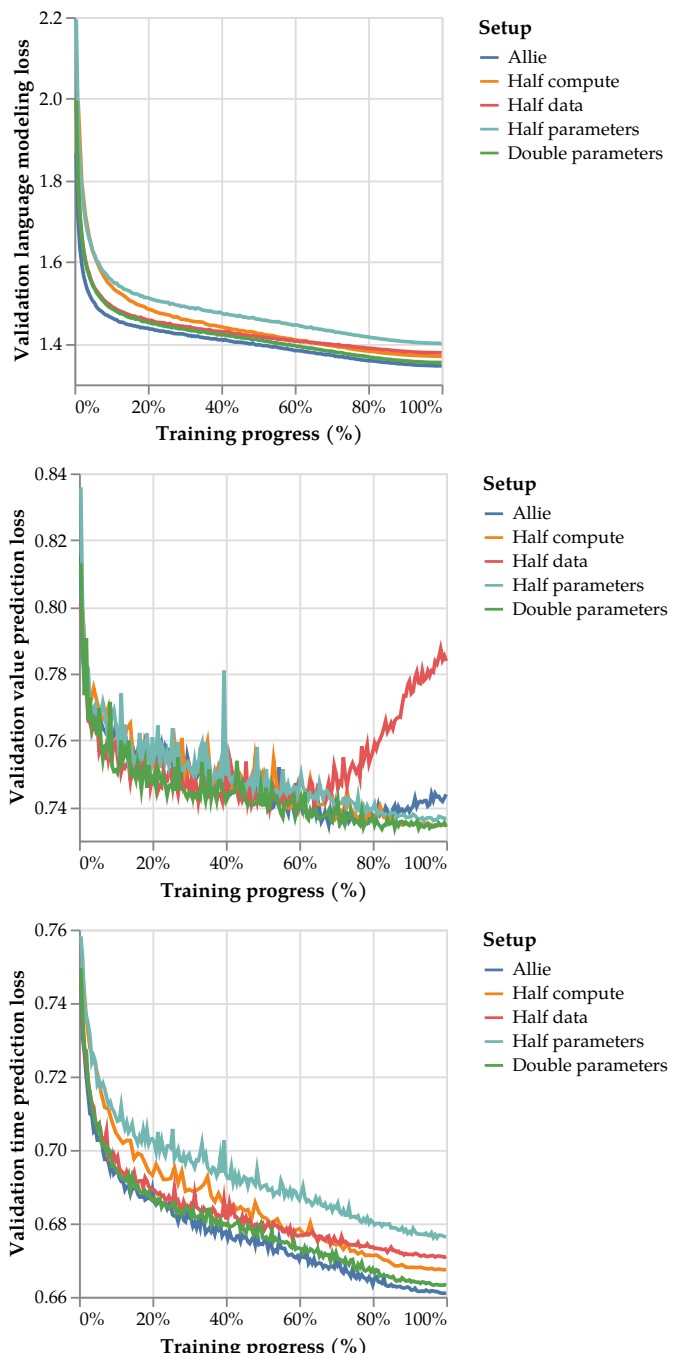

Figure 13: Validation losses of ALLIE and ablations throughout training. **Top**: language modeling loss. **Middle**: value prediction loss. **Bottom**: time prediction loss.

