# OpenReview forum: "Human-Aligned Chess With a Bit of Search"
_ICLR.cc/2025/Conference — ICLR 2025 Poster_

### Official Review · Reviewer_wEKh · 2024-10-24

**Soundness:** 3
**Presentation:** 3
**Contribution:** 2
**Rating:** 6
**Confidence:** 3

**Summary:**

This paper presents ALLIE, a skill-calibrated and human-aligned chess AI trained purely from human games. ALLIE uses the transformer architecture for sequence modelling to predict the distribution of the next move, the pondering time humans spend at the current position, and the winner of the game. Through a test against a database of human games and an online user study, the authors find ALLIE better at predicting human moves than its counterparts. They also propose an adaptive Monte Carlo Tree Search (MCTS) mechanism at inference time, leveraging the predicted pondering time to model human behaviours better and calibrate the skill.

**Strengths:**

- The objectives of the paper are clearly defined, and the paper sticks close to these objectives.
- The writing and presentation of the paper are clear and of good quality. I do not spot any visible typos in the main text.
- Though I am not entirely sure, I believe modelling the pondering time and resignation and utilizing the time prediction for adaptive MCTS are novel ideas in this line of work. In addition, the invention of soft tokens for strength conditioning is a fresh idea.
- The presented evaluation methods and metrics seem rigorous. Furthermore, this work complements the offline human alignment evaluation with surveys consisting of quantitative and qualitative feedback from online users.

**Weaknesses:**

- The design of ALLIE is akin to AlphaGo [1] without the self-play phase. Therefore, I do not view training the neural networks entirely on human data and applying MCTS for policy improvement as an algorithmic advancement.
- The authors stress the necessity of adaptive search in skill calibration. However, failure to match the grandmaster level can be interpreted merely as a lack of strength, and incorporating a planning phase improves ALLIE's strength to close that gap. The authors do not explain why adaptive search attains a better skill calibration error even against mid-level opponents.
- As the authors point out, the users knew they were playing against an AI before they filled out the survey. Therefore, the human alignment result from the user study might not be as objective as if they conducted a Turing test.
- It would better support the superiority of adaptive search in skill calibration if ALLIE with adaptive search were compared against an AlphaZero-like agent, where the skill level is adjusted using a temperature parameter.

[1] Silver, D., Huang, A., Maddison, C. et al. Mastering the game of Go with deep neural networks and tree search. Nature 529, 484–489 (2016). https://doi.org/10.1038/nature16961

**Questions:**

- I wonder how the hyperparameter $c$ for deciding $N_{sim}$ was selected. Was it determined via trial and error?
- Could you explain why the move-matching accuracy is a good metric for human alignment evaluation? For instance, I can imagine that amateur players can make blunder moves in many different ways given the same board position. Thus, ALLIE may get a low move-matching accuracy even with strength conditioning, while the human player feels their opponent is human-like. Continuing with this point, I believe chess beginners are less likely to tell if their opponent is a human simply because they see fewer games and don't have the same level of experience. Therefore, I believe it would be interesting to report the human alignment survey grouped by Elo ratings.

---

> ### Author Response · Authors · 2024-11-21
> **Part 1**
>
> We appreciate your review and constructive feedback, and we are grateful that you consider our writing & presentation clear, and our evaluation rigorous.
>
> > The design of ALLIE is akin to AlphaGo without the self-play phase
>
> The key contribution of our work does not lie in the use of AlphaGo-style MCTS itself, but rather, we show that the use of adaptive search enables chess gameplay at a Grandmaster level (>2500 Elo) without loss of “human-likeness” — which to the best of our knowledge is not possible with prior techniques.
>
> Like the reviewer pointed out, our work 1) shows machines can mimic the depth of human reasoning by learning to predict the think time of humans, and 2) proposes an time-adaptive MCTS procedure that enables strong gameplay and skill calibration, which are quite different from the AlphaGo line of research.
>
>
> > The authors do not explain why adaptive search attains a better skill calibration error even against mid-level opponents.
>
> Adaptive search leads to better skill calibration than the no-search baseline (Allie-policy) because search itself is essential to avoid blunders. Even against players with Elo ~ 1700, baselines that don’t do search underperform humans by > 50 Elo points, and both regular and adaptive search help close this gap and leads to much better calibration.
>
> > As the authors point out, the users knew they were playing against an AI before they filled out the survey. Therefore, the human alignment result from the user study might not be as objective as if they conducted a Turing test.
>
> We acknowledge that the knowledge of playing against a bot may subtly bias behavior, but we don’t expect drastic differences in playing strength or style. In addition, all systems (Maia, Allie-Policy, Allie-Adaptive-Search, etc.) are deployed identically, which should minimize the impact of this knowledge on the validity of our findings. We further note that the terms of service of Lichess (the website we deployed our chess bot on) require disclosure if playing using an engine, and it would be simply impossible to evaluate against thousands of human players if we didn’t deploy our model on Lichess. We would love to see a Chess Turing test study done in the future, which evaluates how well Allie and other chessbots can fool human players.
>
> > It would better support the superiority of adaptive search in skill calibration if ALLIE with adaptive search were compared against an AlphaZero-like agent, where the skill level is adjusted using a temperature parameter.
>
> We certainly could “water-down” any strong chess AI like AlphaZero to achieve perfect skill calibration (e.g., with a temperature parameter, or making it play a random move with some probability p depending on the strength level), but we argue such skill calibration is uninteresting in itself. The core challenge that we try to solve in this paper is to train human-like chess AI models that are well-calibrated to actual humans at varying strength levels, and this is not possible through sampling from chess engines such as AlphaZero.
>
> For example, the Maia paper [1] reports direct comparisons against Stockfish and Leela (an open source version of AlphaZero), and shows that neither engine matches human behavior well (Figure 4).
>
> [1] Aligning Superhuman AI with Human Behavior: Chess as a Model System. Reid McIlroy-Young, Siddhartha Sen, Jon Kleinberg, and Ashton Anderson.

---

> > ### Author Response · Authors · 2024-11-21
> > **Part 2**
> >
> > > I wonder how the hyperparameter $c$ for deciding was selected. Was it determined via trial and error?
> >
> > $c$ is chosen such that we perform 50 rollouts on average across all positions, based on statistics computed on the training dataset. We do 50 rollouts for both adaptive-search and standard MCTS in line with prior work [1] which studied augmenting behavior-cloned chess and go policies with traditional MCTS.
> >
> >  > Could you explain why the move-matching accuracy is a good metric for human alignment evaluation?
> >
> > When move-matching accuracy is high, the actual move made by the model agrees with humans more. Suppose there exists a model with an 100% move-matching accuracy, we argue that it is a perfect human-aligned model, since it always makes the same move as a human would! 100% accuracy is impossible in reality, since human chess play is a stochastic process, but nevertheless higher accuracy means the model behaves more similarly to humans, regardless of the strength level.
> >
> > [1] Modeling Strong and Human-Like Gameplay with KL-Regularized Search. Athul Paul Jacob, David J. Wu, Gabriele Farina, Adam Lerer, Hengyuan Hu, Anton Bakhtin, Jacob Andreas, and Noam Brown.

---

### Official Review · Reviewer_rNsm · 2024-11-01

**Soundness:** 3
**Presentation:** 3
**Contribution:** 2
**Rating:** 6
**Confidence:** 4

**Summary:**

This paper introduces ALLIE, a chess AI trained exclusively in human chess games with the goal of creating a skill-calibrated and human-aligned chess AI. If the policy is solely employed for matches with humans, the AI struggles to align with human players of corresponding Elo ratings. To address this challenge, ALLIE takes into account the pondering time of each move recorded in human games and proposes a time-adaptive Monte Carlo tree search to mitigate the skill gap, which outperforms search-free and standard MCTS baselines. Furthermore, ALLIE also considers human resignation behavior; analyses show that ALLIE achieves an 86.4% true positive rate, indicating that it demonstrates more human-like behavior.

**Strengths:**

This work proposes some novel ideas for developing human-like agents by considering not only move accuracy but also non-move behaviors of human players, such as pondering time and when to resign. It helps bridge the gap in developing human-aligned AIs, as strong AI agents may exhibit non-interpretable behaviors. Additionally, it compares favorably with state-of-the-art agents, MAIA and GPT-3.5, achieving better results.

**Weaknesses:**

## Major Comments:
First, the novelty of this paper is very limited. Specifically, it directly applies the transformer architecture to chess games. The only novel thing is that the approach introduces some new outputs, such as predicting pondering time or resignation. Moreover, for the network architecture, the reason why the transformer architecture is better than MAIA, which uses residual networks, is not explicitly explained. For example, if the parameter size is much larger than MAIA, the inference time might be also much longer, and the fairness of the experiments should be reconsidered. It is better to compare the inference time and the number of parameters under the same hardware.

Second, the evaluation set excludes moves made under time pressure (when the remaining time is less than 30 seconds) to avoid situations where players tend to make random moves. However, these blitz moves under time pressure might reflect human intuitions. Although MAIA’s evaluation also excludes these moves, but since this paper aims to create human-like AIs, it is recommended to include these moves in the evaluation, as they might yield interesting and more convincing results. Furthermore, while non-move behaviors consider the pondering time for each move, human players often decide the thinking time for the next move according to the remaining time. How about incorporating the remaining time as input to models? By doing this, the method might not need to exclude moves made under time pressure in the evaluation set.

Third, for the generality of the proposed method, the experiment in this paper is only applied to chess, it is not sure whether this method can be generalized to other games. The author should include other games to demonstrate the effectiveness of their approach.

## Minor Comments:
- In abstract, “... resignations In offline evaluations,” -> “... resignations in offline evaluations,”
- The time spent for each move should be recorded in the game history, but not explicitly stated in the main text, please address it.

**Questions:**

1. Please address the issues mentioned in the Weakness, especially major comments.
2. Why do transformer-based models perform better than MAIA? Is this due to the architecture itself or the parameter size? Does the transformer-based model in ALLIE have a longer inference time than MAIA*?
3. Can this method be applied to other games or domains? Any further experiments on other games?
4. Do you enable resignation for ALLIE as mentioned in Section 5.2 for the LARGE-SCALE HUMAN STUDY?
5. Do ALLIE-SEARCH and ALLIE-ADAPTIVE-SEARCH use the softmax function for final move decisions, as ALLIE-policy does? If not, the experiment may be unfair.
6. Is the bin range in Figure 5 set to 200? The datasets are separated every 100 Elo ratings. Why does the bin range differ?
7. In section 4.3, it claims "Due to dependency on the textual PGN notation, this approach
is not compatible with OpenAI’s latest chat-based LLMs (e.g., GPT-4)". However, the explanation of why we cannot use GPT-4 for evaluation is unclear.
8. Could you incorporate the remaining time as input to the model? Maybe it might make the agent more human-like.
9. How to determine the constant c in the time-adaptive MCTS procedure? It seems it is not explicitly stated in the main text.
10. For the resignation conditions, you consider both the resignation token and values. What if it only thinks of resign tokens? Would the result be better or worse than TPR of 86.4%?
11. In Maia’s paper, they state that MCTS tends to degrade move prediction performance in their setting. Why does MCTS improve move accuracy, especially in expert-level human games in this work?

---

> ### Author Response · Authors · 2024-11-21
> **Part 1**
>
> Thank you for your detailed read and feedback on our paper. We address some of your concerns below.
>
> > the novelty of this paper is very limited
>
> The reviewer may have missed the main contribution of our work—we combine a policy learned from real human games with an adaptive search. The adaptive search method is a new (and more effective) way of scaling inference-time compute compared to traditional MCTS. In reviewer onxn’s words: “This time-adaptive search proves to be crucial for closing the skill gap between Allie and the human opponent, as shown in the human study the authors conducted.”
>
> > Allie uses a Transformer architecture with many more parameters than Maia
>
> We first want to highlight that our main contribution, the time-adaptive search procedure, is the key algorithmic innovation that enables Allie to play at a grandmaster level (2528 Elo) against 2500 Elo human opponents, while no-search Allie baseline performs much worse (2136 Elo). This gain is independent of the model architecture and parameters.
>
> That said, we completely acknowledge that using a much larger Transformer architecture, which uses more parameters and inference compute plays a role in the improvement of move-matching accuracy over Maia, but Maia is the best open-source chess model that we have access to, and we simply don’t have access to a Maia-like model with equal parameters to compare against. To test the impact of model size on the performance of Allie, we introduced an Allie variant with half the parameters in our ablation study (Appendix F, Figure 9), and the move-matching accuracy (54.2% with half parameters) still convincingly outperformed the Maia model (51.6%). To provide more evidence that model architecture and parameter count clearly isn’t the entire story: when comparing Allie’s move-matching accuracy against GPT-3.5 – a much larger Transformer model that was partly trained on chess data, Allie predicts human moves more accurately on average.

---

> ### Author Response · Authors · 2024-11-21
> **Part 2**
>
> > Evaluation set excludes moves made under time pressure
>
> We acknowledge that “blunders” under time pressure is an interesting behavior to learn, but we decided to follow the experimental setup of Maia to make our training procedure consistent with the previous state-of-the-art method, so our results are more comparable. We agree that adding time remaining for each move can further improve the ability of the model to model human pondering time, this would be a relatively small change and we don’t expect it to change our results significantly.
>
> > Should show results on other games to demonstrate generalizability
>
> Chess is a longstanding challenge in AI development, dating to the 1960s. We strongly believe that contributions to chess AI are meaningful in their own right, without the need to demonstrate generalizability. However, we do think the generalizability of our method is a very interesting question. We expect techniques that lead to more human-aligned chess play would transfer well to other two-player combinatorial games, and could potentially prove useful for domains in which machine learning models are trained to model human-like decision making. We look forward to follow-up work studying these questions.
>
> > Do you enable resignation for ALLIE as mentioned in Section 5.2 for the LARGE-SCALE HUMAN STUDY?
>
> Yes, resignation is enabled.
>
> > Do ALLIE-SEARCH and ALLIE-ADAPTIVE-SEARCH use the softmax function for final move decisions, as ALLIE-policy does? If not, the experiment may be unfair.
>
> Both search variants use softmax to pick a move like Allie-policy.

---

> ### Author Response · Authors · 2024-11-21
> **Part 3**
>
> > Is the bin range in Figure 5 set to 200? The datasets are separated every 100 Elo ratings. Why does the bin range differ?
>
> We chose the bins so we had at least 50 human games in each bucket. Using a small bin range would lead to fewer games in each bucket and more noisy system performance estimates. This choice is independent from how we processed the pre-training dataset.
>
> > the explanation of why we cannot use GPT-4 for evaluation is unclear.
>
> Getting an OpenAI model to play chess requires a non-chat model (specifically, gpt-3.5-turbo-instruct, the latest non-chat model from OpenAI) for feeding in a chess game record (in PGN format), and converting next move prediction to text generation.
> Assistant models such as GPT-4o cannot be used in this way, and does not seem to exhibit the same ability to play chess.
>
> > How to determine the constant c in the time-adaptive MCTS procedure? It seems it is not explicitly stated in the main text.
>
> We follow AlphaZero’s original implementation and use c=1.25. We have clarified this choice in the revision.
>
> > For the resignation conditions, you consider both the resignation token and values. What if it only thinks of resign tokens? Would the result be better or worse than TPR of 86.4%?
>
> The true positive rate would be strictly higher, since Allie only resigns if both of the following conditions are true:
> [RESIGN] is the most likely token
> Board value below a threshold (very close to losing).
> If we take away condition 2, Allie resigns strictly more often, and we would end up with a higher TPR.
>
> > Why does MCTS improve move accuracy, especially in expert-level human games in this work?
>
> Chess is a game fundamentally about computation and search, and it is plausible that expert-level players can search sufficiently deeply such that next token prediction (or more generally, methods that use finite compute) just cannot simulate this computation. In such cases, we can reasonably expect search-based methods to better model expert behavior.

---

> > ### Comment · Reviewer_rNsm · 2024-11-26
> >
> > Thank you for your response, which addresses my concerns. I have adjusted my scores accordingly.

---

> > > ### Author Response · Authors · 2024-12-03
> > >
> > > Thank you for acknowledging our response.

---

### Official Review · Reviewer_aMwP · 2024-11-04

**Soundness:** 3
**Presentation:** 3
**Contribution:** 3
**Rating:** 6
**Confidence:** 3

**Summary:**

This paper focuses on proposing a chess-playing AI (called ALLIE) that is aligned to human gameplay. It does so by training a Transformer-based network on a dataset of human games, taking into account metadata such as the player's skills (Elo score) and their pondering time. By assuming players ponder on critical situations, the authors propose to add a MCTS-based search component proportional to the pondering time. ALLIE's skill level is estimated against numerous human players, and is shown to have an average skill gap of 49 Elo.

**Strengths:**

- Shows strong and accurate performance across a wide range of Elo scores
- Strong performance compared to Maia, which is also human aligned
- Thorough analysis on different aspects of the algorithm (accuracy of move prediction, accuracy of pondering times, accuracy of value estimate)
- Extensive human study

**Weaknesses:**

- Entirely based on offline data, based of recording of human gameplays. This limits the scope of this method for human-aligned AI. Moreover, large parts of the data have been removed for training (downsampling low-Elo games, removing transitions under time pressure), so it seems training ALLIE-like agents require a specific training regimen.
- There does not seem to be any related work on skill calibration. Although I am not an expert in that area, I would be curious about what works have been done in that regard.
- I find the claim that ALLIE has a "reliable board value estimate" a bit misleading. Fig3 (with a Pearson's $r=0.697$) is already quite noisy, and the vast majority of Fig4 is below that number (around 5+ moves from terminal), with 30+ moves from terminal having a weakly correlated $r<0.3$. What would these values be in terms of mean absolute error?
- GPT3.5 has a comparative move-matching accuracy as ALLIE on higher-Elo games, despite having not been explicitly trained for chess. Although I do not expect it to be included in the human evaluation, it would have been interesting to provide a more in-depth comparison between GPT3.5 and ALLIE. Would a fine-tuned GPT3.5 on the same dataset as ALLIE (but in PGN format) result in a human-aligned agent?

**Questions:**

- ALLIE has been trained exclusively on Blitz games. How does this impact the importance of search compared to standard games? I would expect that Blitz players play mostly instinctively. Blitz might thus be enough to train an agent solely on a human dataset of gameplays, but this might be insufficient for standard games. Do you have any insights as to the types of changes that would be required to scale ALLIE to standard games?
- Related to the previous question, the authors mention the need to generalize to out-of-distribution positions. How well would ALLIE (trained on Blitz) perform in standard games?
- Is the Elo score used throughout this paper specific for Blitz? Or is it the player's global Elo score? I would assume the former, but one survey respondent (Table 9, second row) has an Elo of 1940 and mentions he or she "doesn't really play Blitz so I can't imagine I'm very good".
- Since ALLIE is trained on a fixed dataset of different levels of experts, this would be suited to an offline reinforcement learning (RL) setting. More specifically, ALLIE use a GPT-2 like architecture, making it related to Decision Transformers (DT) [1] (execpt of course the search component). Offline RL approaches provide performance and out-of-distribution gains compared to the behavior-cloning-based approach taken by ALLIE. Are there any reasons why the authors chose to behavior cloning?


[1] Chen, L., Lu, K., Rajeswaran, A., Lee, K., Grover, A., Laskin, M., ... & Mordatch, I. (2021). Decision transformer: Reinforcement learning via sequence modeling. Advances in neural information processing systems, 34, 15084-15097.

---

> ### Author Response · Authors · 2024-11-21
> **Part 1**
>
> Thank you for your detailed review of our work. We hope that our response below addresses your questions.
>
> > Entirely based on offline data, based off recordings of human gameplays
>
> We argue that to train a humanlike chess model, we have to rely on offline human data as the main ingredient: as an analogy, you can’t really teach a language model to generate human language without records of how humans write. While it is plausible that additional self-training could improve the strength of our models, it is not clear such methods would help with our objective of creating a “human-aligned” model, since we are not after creating a “superhuman” chess AI.
>
> > Moreover, large parts of the data have been removed for training (downsampling low-Elo games, removing transitions under time pressure), so it seems training ALLIE-like agents require a specific training regimen.
>
> In terms of downsampling low Elo games and pre-processing, we would like to point out that similar data pre-processing steps are standard in model training. For example [OLMo](https://arxiv.org/abs/2402.00838) uses text quality and deduplication filters, and [ResNet](https://arxiv.org/abs/1512.03385) uses random crops for image augmentation. Removing moves under time pressure and downsampling low Elo games are similar in spirit to standard practice, and they should not limit the generality of our approach.
>
> > There does not seem to be any related work on skill calibration. Although I am not an expert in that area, I would be curious about what works have been done in that regard.
>
> We agree that skill calibration is an understudied area and there doesn’t seem to be many papers on this. One very recent paper is [Maia-2](https://arxiv.org/abs/2409.20553) that adds skill embeddings into the neural network, and reports better move prediction compared to the original Maia models, but it is difficult to directly assess its skill calibration, since that would require evaluation against actual humans players.
>
> > I find the claim that ALLIE has a "reliable board value estimate" a bit misleading.
> The reviewer may have misinterpreted the results on time prediction (Figure 3) for value prediction.
>
> Low correlations (<0.3) of Allie’s value prediction with game outcomes when the game is 30+ moves from being finished is very much expected: the outcome of a chess game is fundamentally difficult to predict, and the uncertainty in the outcome decreases naturally as the game progresses (e.g., one of the two player usually gains a decisive advantage). Allie performs almost as well as stockfish (the state-of-the-art chess engine) in predicting the player who wins a chess game at intermediate positions, which we consider to be quite impressive.
>
> In addition, we would like to point out that learning which player would win a chess game is not our main contribution, and we simply need a reasonable value function to guide the time-adaptive MCTS procedure.
>
> > Would a fine-tuned GPT3.5 on the same dataset as ALLIE (but in PGN format) result in a human-aligned agent?
>
> In principle, GPT-3.5 could be fine-tuned to match human behavior, and it would probably outperform Allie. However, there are two reasons for not doing doing this.
> - Limited scientific insight since we don’t know what into the pre-training data of this model in order to give it Chess-playing capabilities.
> - No longer possible to do since OpenAI deprecated finetuning its non-instruction-tuned models. More recent instruction-following models such as gpt-4o don’t seem to exhibit the same strong chess ability as gpt-3.5.

---

> > ### Author Response · Authors · 2024-11-21
> > **Part 2**
> >
> > > ALLIE has been trained exclusively on Blitz games. How does this impact the importance of search compared to standard games? How well would ALLIE (trained on Blitz) perform in standard games?
> >
> > This is a great question — we expect search to be even more important for rapid and standard games (with more thinking time available for players), as humans plan for longer and more deeply in those games instead of relying on intuition. Since most games played on the internet are blitz games, this makes both training and evaluation on longer chess games difficult. However, we encourage future work to explore using search to imitate human chess behavior in longer time controls.
> >
> > > Is the Elo score used throughout this paper specific for Blitz?
> >
> > Yes, the Elo scores are specific to blitz.
> >
> > > Since ALLIE is trained on a fixed dataset of different levels of experts, this would be suited to an offline reinforcement learning (RL) setting. So why not offline RL?
> >
> > Thanks for making the connection between our work and offline RL. One key distinction is that our goal of building a “human-aligned” chess model is different from the standard reward (winning) maximization objective of RL methods (such as Decision Transformer): a superhuman chess bot like AlphaZero would be neither humanlike nor skill-calibrated. In particular, Decision Transformer uses reward conditioning to pick a high-reward action (chess move in our setting), regardless of whether the action is plausible according to the dataset (human chess games). Therefore, a behavior cloning approach (conditioned on the skill level) is particularly well suited for our problem, because we would like the model to imitate, rather than to surpass humans.

---

> > > ### Comment · Reviewer_aMwP · 2024-11-25
> > >
> > > Thank you for your thorough reply. I appreciate your comment on the training data and blitz games. I am still not convinced about the choice of using Pearson correlation to estimate the value function instead of e.g. the absolute error between the predicted outcome of the game and the actual outcome, though I suppose it is sufficient as a sanity check.
> > >
> > > > Decision Transformer uses reward conditioning to pick a high-reward action
> > >
> > > I would argue that, precisely because it uses reward conditioning, you could condition the policy on arbitrary rewards (be it high or low), such that the selected action matches the skill of the human player.
> > >
> > > Don't get me wrong, overall I like the paper, I simply believe the authors should not be so quick to disregard offline RL which, compared to BC, allows to be more robust to trajectories outside of the training data.

---

> > > > ### Author Response · Authors · 2024-12-03
> > > >
> > > > Thank you for your thoughtful comment regarding the idea of conditioning a Decision Transformer with carefully chosen rewards to achieve human-comparable performance.
> > > > One potential challenge with this approach is that even if the model is able to perfectly associate moves with conditioning reward values, picking the right reward values is likely non-trivial.
> > > >
> > > > Consider, for example, a straightforward implementation that conditions the model on zero terminal reward (essentially playing for a draw).
> > > > While this might achieve good calibration, it would result in a "rubberband AI" system that blunders in winning positions and plays superhuman moves in losing positions—behavior that differs significantly from typical human chess play.
> > > >
> > > > In our work, we introduce time-adaptive MCTS as a novel method for improving AI skill calibration, and as you pointed out, enables strong and human-like gameplay as demonstrated by an extensive user study.
> > > > While offline RL methods clearly hold promise in this domain, we believe our current approach provides an effective solution to the skill calibration challenge. We look forward to exploring additional techniques as complementary directions in future work.

---

### Official Review · Reviewer_t49A · 2024-11-08

**Soundness:** 4
**Presentation:** 4
**Contribution:** 4
**Rating:** 8
**Confidence:** 4

**Summary:**

The paper presents a new human-like chess model that exceeds the current SOTA while adding ponder. The authors look at the model compared to the previous SOTA tests and add some new ones, along with user studies to strongly argue for their model's performance.

**Strengths:**

I think this is a good paper presenting an original approach to human-like play modeling and it answered most of my questions in the text.

# originality

The idea of using a transformer to solve chess predates the current LLM craze (I can find a cite if needed), but this the most successful one so far and shows a real improvement compared to the current slate of LLMs for chess papers. Adding the ponder time is an obvious step from a FEN based dataset, but again is done very well. So while most of the ideas are not novel the combination and execution are.

# quality

The experiments look to be done well and the paper flows well. Adding the human studies was good move to really show it's an improvement over Maia.

# clarity

I found figure 3 difficult to read, the median is almost impossible to see. I also don't know what the squares in figure 1b mean from the caption, is it supposed to be a chess board?

# significance

Chess AI has had a resurgence in the last few years and this could aid that work, it also shows a path for many other turned based games. So I think this has a chance of being significant.

**Weaknesses:**

I don't think the term _human-aligned_ is good in this paper. An aligned AI system is already implicitly aligned to humans, but the term does not mean that it acts like a human, just that it works in the best interest of humans. I think a different term should be used for this concept to prevent it from being interpreted in the more common AI alignment sense.

I'm not clear how the <resign> token can be used in practice, even with a false positive rate of 0.1%, that would be ~2% over 20 moves, and only takes one bad resign to distribute a game. Was the resign prediction used in testing the models in the user studies? And how does it look in a real gameplay scenario, not just against single game states?

The lack of code right now is also concerning, my rating is contingent on the authors fulfilling their promises in the text.

**Questions:**

See the Weaknesses section

---

> ### Author Response · Authors · 2024-11-21
>
> We appreciate your positive feedback.
>
> > I don't think the term human-aligned is good in this paper.
>
> We agree that the term “alignment” is typically used to describe “being helpful and harmless to humans”, but we define human-aligned as being both “humanlike” and “skill-calibrated” (see our introduction). Hopefully this is unambiguous in the context of chess.
> > I'm not clear how the <resign> token can be used in practice, even with a false positive rate of 0.1%.
>
> You might have misinterpreted our results. The average game has about 35 moves, and with a false positive rate of 0.1% per move, the model would incorrectly resign once in 28.5 games. When deployed online, Allie resigns when down substantial material or when an opponent is about to promote a pawn (two FENs of positions where Allie resigns *against humans* are provided below), similarly to how humans would resign.
>
> ```
> 1N6/5pk1/1R2n1b1/R5pp/3Pp3/4P1P1/5PP1/4N1K1 b - - 0 38
> ```
>
> ```
> 3k4/pp2RP2/3p4/6B1/3P4/2Pr4/Pn4K1/8 b - - 0 34
> ```
>
> > No code available
>
> Our code is already publicly available on Github, and our models are already available for download on HuggingFace. We have uploaded a zipfile that contains an anonymized version of our code.

---

### Official Review · Reviewer_onXn · 2024-11-09

**Soundness:** 3
**Presentation:** 2
**Contribution:** 3
**Rating:** 8
**Confidence:** 5

**Summary:**

This paper presents Allie, a chess-playing AI model designed to imitate human behavior across a spectrum of skill levels. Allie is a transformer-based model trained on human games from Lichess that, given a board position and the Elo strength of the two players, predicts the next move the player will play (policy head), the think time, and the game outcome (value head). During inference time, Allie uses these prediction heads to add some MCTS search, where the search rollouts are guided by the (human-like) policy and value heads and the number of rollouts is proportional to the (human-like) think time. The main prior work in this area is the Maia project by McIlroy-Young et al., which also trained on human games to create models at different skill levels (KDD '20) and at the individual level (NeurIPS '21, KDD '22). Allie differs from Maia in three main ways: (1) it is a larger, transformer-based model that represents chess positions as a sequence of move tokens; (2) it models think time and resignation; and (3) it adds MCTS search during inference based on the human-like predictions.

Allie achieves ~4% higher move-matching accuracy on average than Maia (a much smaller model) and ~2% higher accuracy than GPT 3.5 (a much bigger model). More importantly, it exhibits better skill calibration across a wide range of Elo ratings (1000 - 2600), as verified by a human study of 7,483 games which shows that Allie incurs a skill gap of only 49 Elo points on average (provided adaptive search is used). Notably, Allie can play at a grandmaster level (2500 Elo) and still retain human-like qualities.

**Strengths:**

There are several aspects of Allie's methodology and results that I find compelling and believe advance the state of the art.

Allie models human chess playing more holistically by also predicting think time and special decisions such as resignation, which were not modeled in prior chess-playing agents work. Granted, these are just additional metadata recorded with Lichess games that prior methods could also train on, but what I found compelling is the authors' observation that these prediction outputs can be used to parameterize an MCTS search. This time-adaptive search proves to be crucial for closing the skill gap between Allie and the human opponent, as shown in the human study the authors conducted.

Allie achieves good move-matching accuracy across a wider range of Elo ratings (Fig. 2), as well as closer skill calibration to human players (Fig. 5), than prior work. In particular, the Maia models achieve good move-matching accuracy (so they feel human-like) but are not as well calibrated to the Elo of human opponents. Additionally, the fact that Allie can play at a grandmaster level (2500 Elo) is an impressive achievement given that behavior cloning on human data typically yields sub-expert policies.

I like that the authors include GPT 3.5 as a challenging baseline. GPT 3.5's move-matching accuracy is remarkable, given that it wasn't necessarily intended to play chess. Allie's performance is still slightly better, but its main advantage is in enabling search (and requiring much fewer parameters).

Allie's architecture is quite sensibly designed. The use of move tokens and history is a natural way of representing a board position for a transformer model, and avoids the need for a visual representation (as used in AlphaZero, Maia, etc.) -- though it is an interesting question if adding such a visual representation would help, given how visually humans process a chess board. I liked the use of "soft tokens" to encode each player's Elo ratings, allowing Allie to condition its output on skill level. I also liked the design of taking the final decoder layer output and parameterizing each prediction using separate linear layers.

The user study was a nice addition to the paper, as it demonstrated Allie's skill calibration in real human games and also allowed the authors to collect survey results on how enjoyable and human-like Allie (and the other baselines) felt.

**Weaknesses:**

There a several things that concern me about the paper's presentation and claims, which I hope can be clarified through the rebuttal.

First, the discussion of Maia in the introduction and related work seems lacking/misleading to me. I don't think it's fair to say that existing systems are not "human-aligned" -- this was precisely the goal of Maia and they did create both skill-level and individual-level (KDD '22) models using the same basic approach (behavior cloning on human data). The individual-level models improve move-matching accuracy over Maia by a similar margin to your improvement over Maia. McIlroy-Young et al. also did some work on style identification (stylometry) that achieved very high accuracy (NeurIPS '21). I suggest citing and mentioning the NeurIPS '21 and KDD '22 papers.

The fact that the published Maia models do not include think time, resignation, or history does not seem that fundamental to me -- there are archived submissions on OpenReview that extend Maia in some of these ways, e.g., deepening the model and achieving move-matching accuracy similar to Allie's. One of these submissions was recently accepted to NeurIPS '24 and should be discussed in this paper: "Maia-2: A Unified Model for Human-AI Alignment in Chess", which presents a unified version of Maia that conditions on the player's Elo rating and achieves better calibration across skill levels. It is worth noting that Allie's architecture, at 355M parameters, is more than an order of magnitude bigger than these Maia models -- you point out size difference as a reason the comparison to GPT 3.5 is unfair, but you do not point out the size difference between Allie and Maia.

Instead of focusing on the above differences, I think a better framing of this paper is to focus on how predicting human-like outputs (including think time) enables you to augment a behavior-cloned model with search, closing the gap between Allie and human players, particularly stronger ones.

It is important to note that adding MCTS search introduces a non-human-like element to your approach. There is not enough evidence to suggest that humans search the game tree in an MCTS-like manner, and in general our understanding of how humans balance board state evaluation with search (calculation) is quite poor. Thus, I would tone down the claims about only relying on human data, since a search algorithm applied to a game tree implicitly encodes non-human information. One recent piece of work that supports your use of think time to limit MCTS is the paper by Russek et al. called "Time spent thinking in online chess reflects the value of computation". It would be helpful if you could report move-matching accuracy as a function of search time; is it the case that when you do more search, move-matching accuracy decreases? Also, how did you choose the constant 'c' in the number of rollouts function? It would be useful to see how sensitive your results (move-matching accuracy, Elo calibration) are to the choice of c.

Related to the above, it's not clear to me why you would expect Allie's value estimates to match those of Stockfish (Fig. 4). Isn't Allie's value head predicting the outcome a *human* would experience if they continued from that board position? I agree that in many situations this would align with Stockfish's assessment (e.g., if the human is up a lot of material, or if it's close to the end game), but it seems more sensible to me to compare Allie's value estimate to the actual outcome of the game, not what Stockfish thinks the outcome should be.

How is it that adaptive search does better than non-adaptive (AlphaZero-like MCTS) search against 2500 Elo players? Is the non-adaptive search time bounded, in addition to being the same for each move?

You observe that using discrete tokens for Elo would be too sparse, but your soft tokens also interpolate between Elos and are represented as tokens. Isn't this representation also sparse, i.e., won't it suffer from a lack of examples associated with each value, or are you representing them differently internally?

The loss function you use doesn't mention normalization or weights to balance the different loss terms. Is this a potential issue, given that these terms capture very different things (e.g., log-likelihood of next move vs. MSA of think time prediction)?

**Update**: Based on our rebuttal discussion and the additional analysis the authors provided as to why Allie's predictive power of human game outcomes differs from Stockfish's, I am increasing my overall score. I do wish the authors had embraced this difference more intentionally, and reframed their paper around it, but what they provided is a good start.

**Questions:**

My questions and suggestions are interspersed in the comments above. I summarize them here for convenience:

1. There is some missing related work. I realize you may not have seen some of these papers in time, but hope they are helpful in revising your paper.
2. The size difference between Allie and Maia's architectures (and its connection to increased move-matching accuracy).
3. The non-human-like aspects of MCTS search. Can you report move-matching accuracy as a function of search time?
4. How sensitive are your results (e.g., move-matching accuracy) to the constant 'c' in the rollouts function?
5. The difference between Allie's value head and Stockfish's evaluation. How accurately does Allie's value head predict game outcomes?
6. Explain why Allie with adaptive search does better than AlphaZero-like MCTS search against 2500 Elo opponents.
7. Are soft tokens encoded in a way that also suffers from sparsity?
8. Any normalization/weighting applied to the loss function terms?

**Details Of Ethics Concerns:**

A user study was done by deploying chess bots online and surveying people afterwards about their experience playing the bots. The deployment of bots on Lichess is a public and controlled process, but it's not clear to me if the deployment and survey collectively require an IRB.

---

> ### Author Response · Authors · 2024-11-21
>
> We really appreciate your comprehensive review of our paper and constructive feedback, both positive and negative. We are encouraged that you find our methodology convincing, and consider our approach as advancing the state of the art.
>
> > There is some missing related work. I realize you may not have seen some of these papers in time, but hope they are helpful in revising your paper.
>
> Thank you for pointing out the relevant papers, including the quite recent Maia-2 paper. Maia-2 and our work share the common goal of better aligning chess AIs with human behavior. We have added a discussion summarizing the similarities and differences among Maia, Maia-2, NeurIPS'21 and KDD'22 and Allie in the revision.
> On the use of the term “human-aligned:” building a human-aligned chess model is precisely the goal of Maia, but we argue that better strength matching with humans can be seen as a component in this alignment, which as you pointed out, we help achieve through a behavior-cloning model with inference-time search. We have revised our language regarding the use of “human-aligned”, particularly in the context of prior work.
>
> > The size difference between Allie and Maia's architectures (and its connection to increased move-matching accuracy).
>
> The size of Allie’s architecture is indeed much larger than Maia’s and we definitely agree it is one of the factors behind the stronger performance of Maia. We have mentioned both the larger size of the training set, as well as additional model parameters as a factor that explains the increase in move-matching accuracy in our revision.
>
> > The difference between Allie's value head and Stockfish's evaluation. How accurately does Allie's value head predict game outcomes?
>
> The reviewer may have misunderstood Figure 4—we are reporting the correlation between Allie’s value predictions and human game outcomes. Since we don’t know if Allie’s predictions are any good if we don’t have a point of comparison, we also computed Stockfish evaluation as an “objective” metric, and show that Allie predicts human game outcomes about as reliably as Stockfish.
>
> > The non-human-like aspects of MCTS search. Can you report move-matching accuracy as a function of search time? / How sensitive are your results (e.g., move-matching accuracy) to the constant 'c' in the rollouts function?
>
> We did not try other search budgets in our experiments, but we agree with the reviewer that this would be an interesting analysis to run—we may explore adding this analysis in our revision in the next few days.
>
> > How did you choose the constant 'c' in the number of rollouts function?
>
> $c$ is chosen such that we perform 50 rollouts on average across all positions, based on statistics computed on the training dataset. We do 50 rollouts for both adaptive-search and standard MCTS in line with prior work [1] which studied augmenting behavior-cloned chess and go policies with MCTS.
>
> > Explain why Allie with adaptive search does better than AlphaZero-like MCTS search against 2500 Elo opponents.
>
> Thanks for bringing [2] to our attention! Adaptive and non-adaptive search use the same amount of compute *on average* throughout a game, and the key difference is that adaptive search allocates more compute to positions that humans spend additional time in. [2] suggests that humans spend more time at positions that search is more valuable. Since Allie’s time predictions are well-correlated with actual human time usage (Figure 3), it is not a complete surprise that doing search proportional to predicted human time usage results in significantly stronger gameplay than uniform allocation of search budget across every position.
>
> > Are soft tokens encoded in a way that also suffers from sparsity?
>
> Since the soft tokens are basically interpolated embeddings based on Elo, they don’t require observing games for every Elo value which helps with the sparsity issue a lot. As an example, suppose that the model never observes games by a 950 Elo player. With interpolated soft tokens, the skill embeddings would still be very close to say that of a 955 Elo player, which we may have training data for.
>
> > Any normalization/weighting applied to the loss function terms?
>
> We normalize labels of pondering time to have variance 1. All three losses (move, value and time prediction) have equal weighting in the training objective. We will clarify this in the revision.
>
>
> [1] Modeling Strong and Human-Like Gameplay with KL-Regularized Search. Athul Paul Jacob, David J. Wu, Gabriele Farina, Adam Lerer, Hengyuan Hu, Anton Bakhtin, Jacob Andreas, and Noam Brown.
>
> [2] Time spent thinking in online chess reflects the value of computation. Evan Russek, Daniel Acosta-Kane, Bas van Opheusden, Marcelo G Mattar, and Tom Griffiths.

---

> > ### Comment · Reviewer_onXn · 2024-11-25
> > **Follow-up to author's response**
> >
> > Thank you for your responses, and for acknowledging and incorporating the related work and other points I raised into your paper.
> >
> > I appreciate the clarification on Fig. 4. I understand that it shows the correlation between Allie's value head and human game outcomes, and comparing it to Stockfish's correlation is certainly useful. However, your comparison of the two does not make as much sense to me. Allie is trained on human games, so it's value head should do a better job of predicting whether a **human** would win the game; in contrast, Stockfish's evaluation is based on heuristics and game tree search, so I would expect it to do worse at predicting human game outcomes. Fig. 4 shows that Allie does better than Stockfish, but your explanation for this is because "it has access to game metadata (in particular, player skill levels) that Stockfish does not". I think it's more fundamental than that, and I think you should dig deeper into the stages of the game where the differences occur. For example, in the endgame Allie and Stockfish have similar predictive power, probably because there are fewer pieces and the advantages are more clear-cut. But in the middle game it's more nuanced: some positions that Stockfish considers a "win" for white/black may not be easy for a human to win; similarly, some positions Allie considers an easy win for a human may be considered drawn/losing by Stockfish. It would be very interesting to explore this more deeply.
> >
> > I appreciate the clarification regarding MCTS's time budget relative to Allie's adaptive search. I now see that Table 1 says MCTS uses 50 rollouts on average; it would be helpful to remind the reader of this, so they understand the comparison more clearly. While I agree that the paper I mentioned [2] makes the case that humans spend more time in "critical" positions where search is more valuable, my main point is that MCTS does not seem human-like to me at all, and we have very little understanding of how **humans** search the game tree. Perhaps your results suggest that how the search is conducted is not as important as the time spent, at least when it comes to high-level metrics like move-matching accuracy. However, I would bet that at higher accuracy levels (e.g., >70%) this fundamental difference will start to matter, and we'll have to do a better job of understanding human search in order to accurately predict their moves.

---

> ### Author Response · Authors · 2024-12-04
> **Responding to Flag for Ethics Review**
>
> > A user study was done by deploying chess bots online and surveying people afterwards about their experience playing the bots. The deployment of bots on Lichess is a public and controlled process, but it's not clear to me if the deployment and survey collectively require an IRB.
>
> Hi, this is the senior author chiming in. The survey we conducted is IRB exempt as per https://www.ecfr.gov/on/2018-07-19/title-45/part-46#p-46.104(d)(2)(ii) .

---

> ### Author Response · Authors · 2024-12-04
> **A deeper analysis of Allie and Stockfish value evaluations**
>
> > Allie considers an easy win for a human may be considered drawn/losing by Stockfish. It would be very interesting to explore this more deeply.
>
> Thank you for your thoughtful comment.
> Allie's supervision on human game outcomes should teach the model to assign high values to positions that a human can convert (as opposed to positions that are theoretically winning under perfect play), and vice versa.
> We initially overlooked fundamental factor that explains the correlation of Allie's value prediction with human game outcomes.
>
> To explore this deeper, we compare the accuracy of Allie's value predictions (vs. Stockfish evaluations) in predicting game outcome at various stages of the game.
> We first filtered for decisive only games (games ending with draws are omitted from the analyses).
>
> **Analysis 1. What is the role of metadata (specifically, player Elo) in value prediction?**
>
> *Table 1: Game outcome prediction accuracy of games with $\le$ 10 Elo gap between players*
>
> | Game Phase | Stockfish | Allie |
> |------------|---|---|
> | Opening | 50.4% | 55.2% |
> | Midgame | 62.9% | 65.6% |
> | Endgame | 73.9% | 74.3% |
>
> *Table 2: Game outcome prediction accuracy of games with $>$ 100 Elo gap between players*
>
> | Game Phase | Stockfish | Allie |
> |------------|---|---|
> | Opening | 65.2% | 76.8% |
> | Midgame | 69.9% | 79.3% |
> | Endgame | 82.5% | 83.7% |
>
> Elo alone contains a lot of information about the likely player to win. In games with a large skill gap (by $\ge$ 100 Elo), Allie outperforms stockfish by 9.4% in middle game in game outcome prediction, which is a lot higher than 2.3% in games with a small skill gap. But like you suggested, Elo is not the full story---in games with almost no skill gap, Allie still outperforms Stockfish in all phases of the game.
>
> **Analysis 2. Does Allie "know" which positions humans can/cannot convert better than Stockfish?**
>
> Following your suggestions, we qualitatively analyze a few randomly chosen positions from our test set where Allie and Stockfish make different predictions of who will win.
>
> In [this position in the opening](https://lichess.org/jRgJgBb1#11), black has a knight for 2 pawns, but is unable to castle. The position is sharp, and although Stockfish gives black a decisive advantage (black has 77.0% probability of winning), Allie prefers white (78.8%). Interestingly, black immediately misplays the position and loses.
> Like you pointed out, Allie does appear to produce better value estimates for such positions that are objectively winning, but difficult for humans to convert.
>
> FEN: `r1bq1b1r/ppp2kpp/2n2n2/3P4/2B5/8/PPPP1PPP/RNBQK2R b KQ - 0 6`
>
> In [this position in the endgame](https://lichess.org/buBaOyJj#100), Allie quite surprisingly considers black winning with about 75% probability when white has an extra queen on the board (stockfish assigns 98% to white winning).
> Our best guess is that Allie inferred that white is low on time by observing the move history---Allie considers black winning starting at [move 34](https://lichess.org/buBaOyJj#67), when white blunders a rook potentially due to being relatively low on time.
>
> FEN: `8/2Q5/p7/6p1/P2P2kp/4P3/4K3/8 w - - 2 51`
>
> However, Allie seems to have blind spots with evaluating checkmates/sacrifices, which strong human players are capable of calculating. Arguably, solving checkmates fundamentally relies on search and is difficult to do with a value function alone. One interesting example is [this position in the middle game](https://lichess.org/fqT5REul#46) between two strong players.
> Allie gives black a 67% probability of winning, but white is objectively winning if they find the queen sacrifice Qe7. The human player finds this move, and wins the game as Stockfish predicted.
>
> FEN: `4r1k1/2p2ppp/p1P2P2/1p4qb/1Q6/1BP5/PP3PpP/3R2K1 w - - 4 24`
>
> We will incorporate these analyses into our paper in the future.

---

### Meta-Review · Area_Chair_sYdE · 2024-12-19

**Metareview:**

The paper stuides human-aligned chess model with a time-adaptive MCTS, and empirically demonstrate that their model exhibits human-like behaviors.  One weakness of the paper is that it does not discuss the non-human-like aspects of MCTS search.

**Additional Comments On Reviewer Discussion:**

In general, reviewers are all positive about this paper. Before rebuttal, the reviewers had concerns about the discrepancy between Allie's ability to predict human game outcomes and stockfish's predictions, and the non-human-like aspects induced by the MCTS search. During the rebuttal, the authors did a deeper dive into the question about the discrepancy between Allie's ability to predict human game outcomes and Stockfish's predictions, which addressed this big concern from the reviewers. Towards the end of the rebuttal, one reviewer is still concerned about the non-human-like aspects induced by the MCTS search, however the reviewer agrees that the paper is strong enough for an acceptance. The reviewer does encourage the authors to further discuss this non-human-like aspects and MCTS in the revision of the paper.

---

### Decision · Program_Chairs · 2025-01-22

Accept (Poster)